# Synthesis and Structure–Activity Relationship Studies of Pyrido [1,2-*e*]Purine-2,4(*1H,3H*)-Dione Derivatives Targeting *Flavin-Dependent Thymidylate Synthase* in *Mycobacterium tuberculosis*

**DOI:** 10.3390/molecules27196216

**Published:** 2022-09-21

**Authors:** Nicolas G. Biteau, Vincent Roy, Cyril Nicolas, Hubert F. Becker, Jean-Christophe Lambry, Hannu Myllykallio, Luigi A. Agrofoglio

**Affiliations:** 1Institute of Organic and Analytical Chemistry, CNRS UMR 7311, Université d’Orléans, Rue de Chartres, CEDEX 2, 45067 Orleans, France; 2Laboratory of Optics and Biosciences, INSERM U 696-CNRS UMR 7645, Ecole Polytechnique, Route de Saclay, CEDEX, 91128 Palaiseau, France; 3Faculté des Sciences et Ingénierie, Sorbonne Université, 75005 Paris, France

**Keywords:** flavin-dependent thymidylate synthase, pyrido[1,2-*e*]purine-2,4(1*H*,3*H*)-dione analogues, structure-activity relationship, sonogashira and Suzuki-Miyaura cross-coupling

## Abstract

In 2002, a new class of thymidylate synthase (TS) involved in the de novo synthesis of dTMP named *Flavin-Dependent Thymidylate Synthase* (*FDTS*) encoded by the *thyX* gene was discovered; *FDTS* is present only in 30% of prokaryote pathogens and not in human pathogens, which makes it an attractive target for the development of new antibacterial agents, especially against multi-resistant pathogens. We report herein the synthesis and structure-activity relationship of a novel series of hitherto unknown *pyrido[1,2-e]purine-2,4(1H,3H)*-dione analogues. Several synthetics efforts were done to optimize regioselective *N^1^*-alkylation through organopalladium cross-coupling. Modelling of potential hits were performed to generate a model of interaction into the active pocket of *FDTS* to understand and guide further synthetic modification. All those compounds were evaluated on an in-house in vitro *NADPH oxidase* assays screening as well as against *Mycobacterium* *tuberculosis ThyX*. The highest inhibition was obtained for compound **23a** with 84.3% at 200 µM without significant cytotoxicity (CC_50_ > 100 μM) on PBM cells.

## 1. Introduction

The excessive use of antibiotics in humans and animals has led to the appearance of multi-resistant bacteria (BMRs) and as a consequence to an increased mortality [1,2]. In 2014, the World Health Organization warned of a risk of antibiotic shortages by 2050 if nothing was done. Today, *E. coli, K. pneumoniae* and *S. aureus* are resistant to more than 50% of the main antibacterial drugs. From a simple natural genetic evolution, the situation has become a global public health problem, promoting the discovery of new therapeutic targets in order to develop new antibacterial substances. In most eubacteria, plants and eukaryotic cells, *thymidylate synthases* (TS or ThyA) [3] provide the only de novo source of 2′-deoxythymidine-5′-monophosphate (dTMP) required for DNA synthesis. The activity of these enzymes is pivotal for bacterial DNA replication and repair. Reductive methylation of 2′-deoxyuridine-5′-monophosphate (dUMP) to 2′-deoxythymidine-5′-monophosphate (dTMP) was catalyzed by three coupling enzymes of folate metabolism [4]. TS catalyzes methylation by using (*6R*)-*N^5^*,*N^10^*-methylene-5,6,7,8-tetrahydrofolate (CH_2_THF) as a methylene and hydride donor, which results in the formation of dTMP and 7,8-dihydrofolate (DHF). *Dihydrofolate reductase* (DHFR) catalyzes reduction of DHF to THF and the s*erinehydroxymethyl transferase* (SHMT) catalyzes the serine-glycine conversion which is concomitant to the conversion of THF to CH_2_THF as cofactors. TS inhibitors were used as cytotoxic agents, but the lack of selective TS bacterial inhibitors over human has hampered their application. More recently, a new class of TS was discovered [5]. This enzyme is encoded by the ThyX gene (formerly Thy1) and is absent in the vast majority of eukaryotic cells and only present in approximately 30% of gram-positive or negative pathogenic prokaryotes [6]. It is a *flavin-dependent thymidylate synthase* (*FDTS*), which has a unique mechanism, structure and gene sequence, making it an attractive new therapeutic target for the development of new selective bioactive compounds. *FDTS* uses a FAD as methylene career intermediate but also as hydride donor through the reduced form FADH_2_ obtained from *NADPH oxidase* activity. Unlike human TS, *FDTS* produces tetrahydrofolate (H_4_folate) indeed of dihydrofolate (H_2_folate) [7]. Thus, *FDTS* can catalyze multiple biotransformation reactions in comparison to the classical TS. So far, few *ThyX* inhibitors have been reported like the 5-fluoro-2′-deoxyuridine-5′-monophosphate (5-FdUMP) and Raltitrexed^®^, but their poor selectivity between *ThyA* and *ThyX* has hampered their development as new drugs [8,9]. The thiazolidine analog **1** (Figure 1)**,** reported by *Myllykalio* et al. [10], exhibited an IC_50_ of 0.057 μM against *ThyX*. The same group reported a second series of *FDTS* inhibitors discovered from high-throughput screening (HTS) of natural and synthetic compounds [11]. After synthetic modifications and biological evaluation, the benzoquinone analogue **2** was reported to have an inhibitory K_i_ value of 28 nM against *ThyX*. Those molecules did not show any mitochondrial toxicity. However, the major matter of discussion is linked to the quinone properties in medicinal chemistry that allow strong redox stress and Michaël acceptor which could conduct to cellular damage and cell protein alkylation, [12]. *C^5^*-modifed dUMP analogs were described by *Herdewijn* et al. [13]; among them, compound **3** exhibited an IC_50_ value of 0.91 μM. More recently, new series of inhibitors bearing benzo[*b*][1,4]oxazin-3(4*H*)-one scaffold was reported by high-throughput screening of commercially available compounds. Structure activity was performed and led to compound B1-PP146 (4) with an IC_50_ value of 0.69 μM against ThyX [14].

*FDTS* is thus an attractive antibacterial target to the development of new and specific drugs to overcome bacterial resistance. By screening of our in house-synthesized compound libraries, two new pyridopurines analogs **5** and **6** were found to exhibit 23.1% and 33.2% *ThyX* inhibitory effect at 200 µM, respectively. Based on those results, we decided to use pyrido [1,2-*e*]purine-2,4(1*H*,3*H*)-dione as scaffold to new and more active compounds through diversity-oriented synthesis. The current study aimed at identifying substituents at *N^1^* and/or *N^3^* positions, which could increase *ThyX* inhibitory activity. In this manuscript, the *FDTS* enzyme from *M. tuberculosis* (*MtbThyX*) was chosen for biological and docking assays due the emergence of *multidrug-resistant* strains of *Mycobacterium tuberculosis (MDR-TB) [15]*.

## 2. Results and Discussions

### 2.1. In Vitro Mycobacterial Thyx Inhibition Assay for Structure–Activity Relationship Studies

To investigate Structure-Activity Relationship (SAR) in a systematic way, we used a *NADPH oxidase* spectrophotometric assay, adapted from Basta et al. [16], to test the in vitro inhibitory activities of synthesized compounds on *Mtb* ThyX, and discuss the influence of substituents. The most active of our compounds was used as reference scaffold for future optimization. Assay reactions, with a final volume of 100 µl, consisted of 750 µM NADPH, 100 µM dUMP, 2 mM MgCl_2_, 1% glycerol, 50 µM FAD and 10 µM of *ThyX*. The reactions were initiated by injection of NADPH (15 µL of a 5 mM solution) to each well of the microtiter 96-well clear flat-bottom plate, followed by rapid shaking of the microplate. ThyX activity was determined by following the decrease of absorbance at λ_340_ (due to oxidation of NADPH). All assays used a kinetic mode of a multilabel microplate reader with an injector. The primary screen was performed with molecules dissolved in DMSO, including DMSO alone as low-activity control. B1-PP146 (**4**), with a 1,4-benzoxazine moiety and described as tight-binding inhibitors, was used as reference compound [17]. All screening reactions were performed in duplicates at 200 µM concentration.

### 2.2. Modification at the N^3^ Position of Purine Scaffold and SAR Studies

Firstly, *N^3^*-position of pyrido [1,2-*e*]purine-2,4(*1H,3H*)-dione scaffold were modified by various *N*-substitutions (arylations, benzylation, etc.). Three different methods were reported to synthesize pyrido [1,2-*e*]purine-2,4(1*H*,3*H*)-dione libraries. Through C–H annulation from anilopyrimidine [18], by a Ugi–Strecker and isocyanate cyclisation [19,20] or by copper catalyzed cyclisation [21]. In order to access to substituted tricyclic structure with *N^3^*-aryl or benzyl diversities, we choose the successive double cyclisation starting from aminopyridine derivatives (Figure 1). Commercially available 2-aminopyridine 7 and ethyl glyoxylate afforded in moderate yield, the intermediate 8 through Ugi–Strecker reaction in presence of trimethylsilyl cyanide and DABCO under microwave irradiation (MW) at 120 °C during 15 min. Treatment of 8 in presence of EtONa and isocyanates, such as phenyliso- benzyliso- or thioisocyanate derivatives under microwave irradiation afforded the corresponding tricyclic scaffolds (**9a–j**), (**10a–d**), **11** and **12**.

The synthesized compounds, pyrido [1,2-*e*]purine-2,4(*1H,3H*)-dione **9a****–****j** and thio derivative **11** substituted at *N^3^*-position by phenyl, were evaluated for their abilities to inhibit the *ThyX* enzyme (Table 1)

To probe into the optimal scaffold of central heterocycle, the phenyl *para* substitution was studied. With small electron-withdrawing groups, such as fluorine (**9a**), chlorine (**9b**) and bromine (**9c**), a 49–59% inhibition was obtained with the best value for the fluorine derivative. By increasing hydrophobic and electron-withdrawing effects (**9e**), less inhibitory activity was observed (31.7%). Substitution at *para* position with electro-donating group (–OCH_3_) **9f** decreases the activity to 8.1% whereas the methyl group (**9d**) led to 55% inhibition. On the other hand, fluorine analogs of **9a**, by substitution at meta position (**9g**) or a second fluorine substitution at *ortho/para* (**9h**) or *meta/para* (**9i**) positions, results in loss of *ThyX* inhibition. Polysubstitution with electron-donating group such as methyl groups (**9j**) also display low enzyme inhibition. We investigate the substitution of oxygen by sulfur atom at 2 position, and we observe that compound **11** decreases drastically the inhibitory effect from 59.4% to 19.8% to compare with **9a**. We also looked at the influence of flexibility induced by benzylic substitution (Table 2). Both benzyl analogs **10a–d** and **12** showed a significant loss of activity (<25%) in comparison with the aryl derivatives **9**.

Overall, this SAR study reveals that flexibility, sulfur substitution, polysubstitution or large groups decrease inhibitory activities. On phenylic substitution, small electron-withdrawing or electron-donating have shown reasonable inhibition (>50%). To determine central scaffold hit, other assays were performed by decreasing [FAD] inhibitors concentration, through tritium release assay (Table 3). Without FAD adding, only 1.1–1.5 of 4 active sites were occupied by this natural cofactor. Without excess of FAD, we can detect a competition effect between FAD and potential inhibitors. By decreasing inhibitors concentration to 50 μM, we can study the true potential of our molecules. The tritium release assay permitted us to study inhibition on the second mechanism part (methylene transfer).

Compound **9a** showed better activity, by increasing inhibition to 76.1% at 200 μM in absence of FAD, which was not the case for compounds **9b** and **9d**, for which the inhibition effect had decreased. With a 50 µM inhibitor concentration, only compound **9a** provided an equal inhibitory activity to the reference compounds **5** and **6**, and no inhibition was observed for **9b** and **9d**. Through a tritium release assay, compounds **9b** and **9d** exhibited increased inhibitory activities on *ThyX* with 77.1% and 69.2% at 200 µM, respectively, which suggested that these molecules would be more active in the methylene transfer mechanism. On the other hand, compound **9a** showed similar inhibition on *Mtb ThyX* (59.5% at 200 μM, Table 3) to the NADPH oxidase assay (59.5% at 200 µM, Table 1). Because most of the active compounds on the first part of the mechanism were also active on the second part (tritium release assay), NADPH oxidase assay was taken as a reference test to determine inhibitory activities and compound **9a** as the scaffold to perform other modifications in search of better activity.

### 2.3. Modification at the N^1^ Position of Compound ***9*** Taken as Scaffold and SAR Studies

Starting from **9a** as scaffold, a library of molecules with structural modifications through *N^1^*-alkylation with various benzyl groups was obtained (**13a–f**) (Figure 2). In the first attempt, in the presence of K_2_CO_3_, in DMF at room temperature during 12 h a competition between N^1^- and O^2^-alkylation was observed (by HMBC-NMR), which decreased the yield and deteriorated the purification. In this case, we investigated the solvent effects (DMF, THF), base influence (K_2_CO_3_, Cs_2_CO_3_, LiH, NaH, etc.,) and activation mode (microwave, sonication) to enhance the *N^1^*-regioisomers. We observed that only the *N^1^*-alkylated regioisomer was obtained under microwave activation at 120 °C during 20 min with potassium carbonate and molecular sieves in presence of various bromobenzyl analogs (Figure 2).

The obtained benzylic derivatives **13a–f** were then evaluated against *FDTS* (Table 4).

We observed that the presence of benzyl group (**13a**), and substituted derivatives at the *para* position by methyl (**13b**) or electron-withdrawing group (**13c–f**) result in lack of inhibition activity compared to the reference **9a** (59.4%). From this study, further modifications at *C^7^* and *C^8^* moiety of compound **9a** with free *N^1^*-position were thus performed.

### 2.4. Modification at the C^7^ and C^8^ position of Compound ***9a*** and SAR Studies

After modulation on the pyrimidine-2,4-dione ring, we investigated the influence of various substituents on the aromatic ring at *C^7^* and *C^8^* positions. We first synthesized under microwave activation two series of substituted molecules by methyl group (**18a,b**) or bromine (**19a,b**) (Figure 3). Halogen derivatives **19a,b** were used as starting material to develop a library of 24 molecules through palladium cross-coupling reactions (Figure 4).

#### 2.4.1. Modifications by Sonogashira Cross-Coupling Reaction

Several substituents were introduced at *C^7^* and *C^8^* positions of **19a,b** under different palladium cross-coupling reactions. Sonogashira conditions [22] were investigated under different activation modes (thermic [23], ultrasonication [24], and microwaves [25]). Under microwave irradiations in DMF with triethylamine, in presence of CuI and Pd(PPh_3_)_4_, the aromatic analogues **20** and **21a,b** and aliphatic alkynes **20** and **21c–g** derivatives were obtained in moderate to good yield (40 to 92%) (Figure 4).

All those compounds were then evaluated (Table 5).

The different substituents on *C^7^* and *C^8^* position were chosen for their abilities to generate flexibility, to create a hydrogen bond, hydrophobic interactions and π-stacking interactions with the active site of the enzyme. Modifications at the *C^7^*-position led to derivatives with aromatic alkynes, which induce flexibility, the presence of aliphatic chain at *para* position (compounds **20a** and **20b** respectively) or aliphatic chain with 5, 6 and 7 (compounds **20c**, **20d** and **20e** respectively) which have shown no to low inhibition still below that of the **9a** activity. Compounds **20f** and **20g**, substituted by side-chain presenting hydrogen-bound site with amide and urea moiety (respectively) and hydrophobic chain, did not show more efficiency against ThyX. The same substituents were introduced at *C^8^*-position of **9a**. Even if most of the molecules do not present an inhibitory activity higher than 33%, this position is however more favorable than that in *C^7^*. In fact, we observe for the aromatic group derivatives **21a** and **21b**, higher inhibitions than the position *C^7^*, and especially for **21b** inhibitory activity was higher (69.1% at 200 μM) than **9a** (59.4%). Overall, this SAR study reveals that the aromatic ring at *C^8^*-position increases inhibitory activities as well as maybe some flexibility. π-stacking interactions seem to be predominant in order to increase the activity, which is why we have synthesized a new series of molecules bearing aromatic groups by the Suzuki-Miyaura cross-coupling reaction. Expected molecules were planar and could be intercalated between the different co-factors into the enzyme pocket.

#### 2.4.2. Modifications by the Suzuki-Miyaura Cross-Coupling Reaction

Modifications under the Suzuki-Miyaura cross-coupling conditions [26] were also performed under microwave irradiations [27,28] in presence of Cs_2_CO_3_, Pd(PPh_3_)_4_, and various boronic acid derivatives in DMF at 120 °C for 40 min. A library of **9a** analogs substituted by aromatic derivatives was isolated (**22**, **23a–e**) in 11 to 50% yield (Figure 5).

All those compounds were then evaluated (Table 6).

The aromatic substitution at *C^7^*-position-bearing with hydrogen or methyl (compound **22a** and **22b** respectively) showed low (**22a**: 20.7%) to moderate (**22b**: 45.7%) inhibition. With electron-withdrawing phenylic substitution such as methoxy, trifluoromethyl ether or fluorine (**22c**, **22d** and **22e** respectively) inhibitory activities do not exceed 28%. At *C^8^* position, the same substitutions were realized with electron-donating substituents at phenylic *para* position (compounds **23a** and **23b**). With **23b** equal inhibition was observed as the *C^7^* position. For **23a** a promising inhibition was observed with 84.3% at 200 µM. For electron-withdrawing substituents at phenylic *para* position, higher inhibitory activities were shown with methoxy (35.4%) and fluorine (43.1%) group (compound **23c** and **23e** respectively). In the case of trifluoromethyl ether **23d**, equal inhibition was observed. Overall, this SAR study revealed that small electron-withdrawing and donating groups showed only less to reasonable inhibitions. One of those compounds with phenyl substitution on the *C^8^* showed the best inhibition of this study at 200 µM. Planar structure and π-stacking interactions looked to be the best characteristic to develop potential *Mycobacterium tuberculosis ThyX* inhibitors.

### 2.5. Mycobacterial ThyX Docking Studies

dUMP was docked into the binding site of ThyX from the *Mycobacterium tuberculosis* complex (Figure 2) with FAD, dUMP (PDB: 3GWC) as the starting structure to perform an initial molecular study. dUMP binds in the *FDTS* pocket by π-stacking with FAD, but also by several hydrogen bonds [29]. On the pyrimidine-2,4-dione ring with four hydrogen bonds: two between *C^2^=O* and Arg199; one between *N^3^* and Arg199; and finally one between *C^4^=O* and Arg107. On the 2′-deoxyribosyl part, only two hydrogen bonds were created between 3′-OH, Arg95 and Gln103. The last hydrogen bonds were localized on the phosphate moiety of dUMP. This part of the natural subtract acted like an anchor into active site; six hydrogen bonds were reported between oxygens of phosphonic group and Arg87 (2 bonds), Gln106 (1 bond), Arg172 (2 bonds) and Arg107 (1 bond).

Following this molecular modeling, compounds **9a** (Figure 3) and **23a** (Figure 4) with the highest inhibitions and results at NADPH oxidase assay and tritium release assay, were docked into the FAD and dUMP pocket with 13 amino acid residues. Compound **9a** (Figure 3) was the first docked into *Mycobacterium tuberculosis* pocket. It showed reasonable inhibition at 200 µM (59.4%), a reduced activity at 50 µM (18.4%). By diminution of [FAD] it showed higher inhibition (76.1%) that suggested possible competition with the natural co-factor. Compound **9a** took place closure than the FAD, its fluorophenyl group near the central ring of FAD, which let assumed π-stacking interactions. On the other hand, 2D representation into the pocket showed several hydrogen bond interactions with amino acid residues. Compound **9a** created five interactions with amino acid responsible of hydrogen bond formation with dUMP. The pyrimidine ring created three interactions with Leu104 and Gln103 on its *N^1^* position. One bond was created with the *C^4^=O* and Arg172. The last two hydrogen bonds were placed on the *N^5^* with Arg172 and on F with Arg199. The loss of inhibition could be explained by the large size of the S atom of compound 11, which then changes the positioning of the substrate in the active site of the protein. Delocalization **9g** and polysubstitution **9h**, **9i** and **9j** changed the capacity to create the hydrogen bond, but also the ring aromaticity. When fluorine was substituted by trifluorimethyl ether **9e**, the capacity to create bond hydrogen was replaced by a hydrophobic interaction, leading to a loss of inhibitory activity. With a smaller group like methyl **9d**, inhibition stays equal to compound **9a**. Hydrophobic interaction with a small group was equal to hydrogen bond interaction in this case. With methoxy substitution **9f**, the loss of inhibition could result from the change of bond angle. Insertion of carbon between the pyrimidinedione ring and phenyl **10a–d**, **12** increase flexibility of the structure and broke the planar scaffold. The loss of inhibition observed on *N1*-alkylated compound **13a–h** could be explained by the loss of the two hydrogen interactions from this position. Compound **23a** (Figure 4) was docked into *Mycobacterium tuberculosis* ThyX pocket. This compound presented the highest inhibition at 200 µM.

Compared to **9a**, compound **23a** looked to be less stabilized into the pocket because of three hydrogen bonds interactions’ loss. It still conserved two major hydrogen bond interactions with Arg95 (hydrogen bond interaction with 3′OH of dUMP) with *C^4^=O* and Arg172 (hydrogen bond with phosphonic acid part of dUMP) with *C^2^=O*. By looking at 3D modeling, we observed three new interactions which could explain biological results. Two π-stacking interactions from the phenyl with Tyr108 and His203. Another π-stacking could be observed on the *p*-fluorophenyl part with His91. The loss of inhibition by decreasing FAD concentration could be explained by the possible π-stacking interaction between isoalloxazine and central ring of pyrido [1,2-*e*]purine-2,4(*1H,3H*)-dione. However, one of the biggest difficulties with molecular docking of weak inhibition was the position into the pocket. Some of them were reproducible but a small modification could change everything; the large flexible pocket of *FDTS* was responsible for this issue.

## 3. Materials and Methods

### 3.1. Chemistry General Section

Commercially available chemicals were provided as reagent grade and used as received. Some reactions requiring anhydrous conditions were carried out using oven-dried glassware and under an atmosphere of dry Argon. All anhydrous solvents were provided from commercial sources as very dry reagents. The reactions were monitored by thin layer chromatography (TLC) analysis using silica gel precoated plates (Kieselgel 60F254, E. Merck). Compounds were visualized by UV irradiation and/or spraying with sulfuric acid (H_2_SO_4_ 5% in ethanol) stain followed by charring at average 150 °C. Flash column chromatography was performed on Silica Gel 60 M (0.040–0.063 mm, E. Merck). The infrared spectra were measured with Perkin–Elmer Spectrometer. The ^1^H and ^13^C NMR spectra were recorded on BrukerAvance DPX 250 or BrukerAvance 400 Spectrometers. Chemical shifts are given in ppm and are referenced to the deuterated solvent signal or to TMS as internal standard and multiplicities are reported as s (singlet), d (doublet), t (triplet), q (quartet) and m (multiplet). Carbon multiplicities were assigned by distortion less enhancement by polarization transfer (DEPT) experiments. ^1^H and ^13^C signals were attributed on the basis of H–H and H–C correlations. High Resolution Mass spectra were performed on a Bruker Q-TOF MaXis spectrometer by the “Fédération de Recherche” ICOA/CBM (FR2708) platform. LC-MS data were acquired on a Thermo-Fisher UHPLC-MSQ system equipped with an electron spray ionization source (ESI). The temperature of the source was maintained at 350 °C. Initially, the cone voltage was set at 35 V and after 5 min was increased to 75V. In full scan mode, data were acquired between 100 and 1000 m/z in the positive mode with a 1.00 S scan time. In addition, a UV detection was performed with a Diode array detector at three wavelengths 273, 254 and 290 nm, respectively. A water/methanol (70%/30%) solution mixture with 0.1% formic acid was used as mobile phase. The composition of the mobile phase was increased to 100% methanol with 0.1% formic acid with a 7% ramp. The flow rate was set at 0.300 mL.min^−1^. Samples diluted in the mobile phase were injected (3 μL) on a C18 column (X-terra, Waters), 2.1 mm internal diameter, and 100 mm length placed into an oven at 40 °C. Electronic extraction of ions was performed and the subsequent areas under the corresponding chromatographic peaks determined. 

### 3.2. General Synthetic Procedure 1 for Strecker-Ugi Cyclization

A mixture of amino pyridine derivatives (200 mg) and ethyl glyoxalate (50% solution in toluene) (1 eq.) was stirred at 25 °C for 2 min. THF (4.5 mL) and 1,4-diazabicyclo [2.2.2] octane (1 eq.) were subsequently added. The reaction mixture was cooled to 0–5 °C and cyanotrimethylsilane (1 eq.) was added. The mixture was heated under microwave irradiation at 120 °C for 15 min. The solvent was evaporated under reduced pressure. The crude product was dissolved with EtOAc, washed with K_2_CO_3_, dried over MgSO_4_, filtrated and concentrated under reduced pressure.

Ethyl 3-aminoimidazo [1,2-*a*]pyridine-2-carboxylate (**8**). The title compound was prepared from commercially available **7** to afford after purification the desired product as a yellow solid (50%). CAS # 1487454-00-1. ^1^H NMR (250 MHz, DMSO-*d_6_*) δ 8.17 (dt, 1H, *J* = 7.1, 1,1 Hz, H_4_), 7.31 (dt, 1H, *J* = 9.3, 1,1 Hz, H_7_), 7.05 (ddd, 1H, *J* = 9.3, 6.5, 1.1 Hz, H_6_), 6.79 (ddd, 1H, *J* = 7.1, 6.5, 1.1 Hz, H_5_), 6.35 (bs, 2H, NH_2_), 4.27 (q, 2H, *J* = 7.1 Hz, O–CH_2_), 1.31 (t, 3H, *J* = 7.1 Hz CH_2_–CH_3_) ppm.

Ethyl 3-amino-7-methylimidazo [1,2-*a*]pyridine-2-carboxylate (**16a**). The title compound was prepared from commercially available **14a** to afford after purification the desired product as an orange solid (45%). CAS # 1216262-11-1. ^1^H NMR (400 MHz, DMSO-*d_6_*) δ 7.98 (s, 1H, H_4_), 7.24 (d, 1H, *J* = 9.4 Hz, H_7_), 6.93 (d, 1H, *J* = 9.4 Hz, H_6_), 6.35 (bs, 2H, NH_2_), 4.26 (q, 2H, *J* = 7.0, Hz, O–CH_2_), 2.22 (s, 3H, C–CH_3_), 1.30 (t, 3H, *J* = 7.0, Hz CH_2_–CH_3_) ppm.

Ethyl 3-amino-6-methylimidazo [1,2-*a*]pyridine-2-carboxylate (**16b**). The title compound was prepared from commercially available **14b** to afford after purification the desired product as an orange solid (34%). CAS # 1498691-03-4. ^1^H NMR (400 MHz, Acetone-*d_6_*) δ 7.62 (d, 1H, *J* = 7.1 Hz, H_4_), 6.62 (s, 1H, H_7_), 6.19 (d, 1H, *J* = 7.1 Hz, H_5_), 5.82 (bs, 2H, NH_2_), 3.80 (q, 2H, *J* = 7.0, Hz, O–CH_2_),1.82 (s, 3H, C–CH_3_), 0.85 (t, 3H, *J* = 7.0, Hz CH_2_–CH_3_) ppm.

Ethyl 3-amino-7-bromoimidazo [1,2-*a*]pyridine-2-carboxylate (**17a**)**.** The title compound was prepared from commercially available **15a** to afford after purification the desired product as a yellow solid (47%). CAS # 1536009-01-4. ^1^H NMR (400 MHz, DMSO-*d*_6_) δ 8.16 (d, 1H, *J* = 7.4 Hz, H_4_), 7.86–7.71 (m, 2H, H_7_), 6.96 (dd, 1H, *J* = 7.4, 2.1 Hz, H_6_), 6.47 (bs, 2H, NH_2_), 4.27 (q, 2H, *J* = 7.1 Hz, O–CH_2_), 1.30 (t, 3H, *J* = 7.1 Hz, CH_2_–CH_3_) ppm.

Ethyl 3-amino-6-bromoimidazo [1,2-*a*]pyridine-2-carboxylate (**17b**). The title compound was prepared from commercially available **15b** to afford after purification the desired product as a yellow solid (44%). CAS # 82193-31-5. ^1^H NMR (250 MHz, DMSO-*d*_6_) δ 8.55 (s, 1H, *J* = 1.8, 0.9 Hz, H_7_), 7.32 (dd, 1H, *J* = 9.7, 0.9 Hz, H_4_), 7.13 (dd, 1H, *J* = 9.7, 1.8 Hz, H_5_), 6.47 (bs, 2H, NH_2_), 4.27 (q, 2H, *J* = 7.1 Hz, O–CH_2_), 1.30 (t, 3H, *J* = 7.1 Hz, CH_2_–CH_3_) ppm.

### 3.3. General Synthetic Procedure 2 for Isocyanate Cyclisation

To a solution of amino ester compound (350 mg) in anhydrous ethanol (4.5 mL), were added subsequently isocyanate (2 eq.) and sodium ethoxide (2 eq.) under inert atmosphere. The reaction mixture was heated at 120 °C under microwave irradiation for 20 min. The solvent was evaporated under vacuum. Pure compounds were obtained after purification by flash column chromatography with DCM/MeOH (92:8) as eluent.

3-(4-Fluorophenyl)pyrido [1,2-*e*]purine-2,4(*1H,3H*)-dione (**9a**). The title compound was prepared from commercially available **8** to afford after purification the desired product as a light yellow solid (75%). CAS # 1842362-44-0. ^1^H NMR (400 MHz, Methanol-*d_4_*) 8.36 (d, 1H, *J* = 7.0 Hz, H_6_), 7.55 (d, 1H, *J* = 9.4 Hz, H_9_), 7.41–7.37 (m, 1H, H_8_), 7.35–7.28 (m, 2H, 2 × H_arom_), 7.26–7.18 (m, 2H, 2 × H_arom_), 7.00 (t, 1H, *J* = 7.0, Hz, H_7_) ppm (see Appendix A).

3-(4-Chlorophenyl)pyrido [1,2-*e*]purine-2,4(*1H,3H*)-dione (**9b**)**.** The title compound was prepared from commercially available **8** to afford after purification the desired product as a light-yellow solid (39%). CAS # 1842362-41-7. ^1^H NMR (250 MHz, DMSO-*d_6_*) δ 8.47 (d, 1H, *J* = 6.9 Hz, H_6_), 7.56–7.48 (m, 3H, H_9_, 2 × H_arom_), 7.34–7.23 (m, 3H, H_8_, 2 × H_arom_), 7.00 (t, 1H, *J* = 6.9 Hz, H_7_) ppm.

3-(4-Bromophenyl)pyrido [1,2-*e*]purine-2,4(*1H,3H*)-dione (**9c**). The title compound was prepared from commercially available **8** to afford after purification the desired product as a white/grey solid (23%). ^1^H NMR (250 MHz, DMSO-*d_6_*) δ 8.54 (d, 1H, *J =* 7.0 Hz, H_6_), 7.70–7.61 (m, 2H, 2 × H_arom_), 7.52 (d, 1H, *J =* 9.4 Hz, H_9_), 7.25 (td, 3H, *J =* 6.2, 2.4 Hz, H_8_, 2 × H_arom_), 6.96 (t, 1H, *J =* 7.0 Hz, H_7_) ppm. ^13^C NMR (63 MHz, DMSO-*d*_6_) δ 158.9 (C=O), 151.7 (C=O), 141.5 (C_quat_), 136.5 (C_quat_), 135.7 (C_quat_), 131.8 (4 × C_arom_), 126.1 (C_arom_), 124.2 (C_arom_), 120.4 (C_quat_), 118.1 (C_arom_), 117.6 (C_quat_), 112.3 (C_arom_) ppm. HRMS-ESI (*m/z*) [M+H]^+^ calcd. for C_15_H_9_BrN_4_O_2_: 356.9981 found: 356.9980. λ_abs_: 263 nm λ_em_: 527nm λ_exc_: 286 nm. R*_f_* 0.43 (DCM/MeOH, 9:1).

3-(*p*-Tolyl)pyrido [1,2-*e*]purine-2,4(*1H,3H*)-dione (**9d**). The title compound was prepared from commercially available **8** to afford after purification the desired product as a light yellow solid (24%). CAS # 1842362-40-6. ^1^H NMR (250 MHz, DMSO-*d_6_*) δ 8.52 (d, 1H, *J* = 7.0 Hz, H_6_), 7.55 (d, 1H, *J* = 9.4 Hz, H_9_), 7.33–7.22 (m, 3H, H_8_, 2 × H_arom_), 7.14 (d, 2H, *J* = 8.2 Hz, 2 × H_arom_), 6.99 (t, 1H, *J* = 7.0 Hz, H_7_), 2.36 (s, 3H, C–CH_3_) ppm.

3-(4-(Trifluoromethyl)phenyl)pyrido [1,2-*e*]purine-2,4(*1H,3H*)-dione (**9e**). The title compound was prepared from commercially available **8** to afford after purification the desired product as a yellow solid (58%). ^1^H NMR (400 MHz, DMSO-*d_6_*) δ 8.38 (d, 1H, *J* = 6.9 Hz, H_6_), 7.80 (d, 2H, *J* = 8.2 Hz, 2 × H_arom_), 7.49–7.39 (m, 3H, H_9_, 2 × H_arom_), 7.23–7.14 (m, 1H, H_8_), 6.85 (t, 1H, *J* = 6.9 Hz, H_7_) ppm. ^13^C NMR (101 MHz, DMSO-*d*_6_) δ 160.1 (C=O), 154.5 (C=O), 143.0 (C_quat_), 141.7 (C_quat_), 131.0 (C_arom_), 127.9 (d, *J_C–F_* = 31.6 Hz, C_quat_), 126.2 (d, *J_C–F_ =* 22.7 Hz, C_arom_), 126.1 (C_quat_), 125.9 (d, *J_C–F_* = 3.7 Hz, 2 × C_arom_), 124.5 (C_arom_), 123.4 (C_quat_), 118.6 (2 × C_arom_), 117.6 (C_quat_), 111.8 (C_arom_) ppm. ^19^F NMR (376 MHz, DMSO-*d_6_*) δ –60.86 ppm. HRMS-ESI (*m/z*) [M+H]^+^ calcd. for C_16_H_9_F_3_N_4_O_2_: 347.0750 found: 347.0751. λ_abs_: 264 nm λ_em_: 486 nm λ_exc_: 266 nm. R*_f_* 0.49 (DCM/MeOH, 9:1).

3-(*p*-Methoxyphenyl)pyrido [1,2-*e*]purine-2,4(*1H,3H*)-dione (**9f**). The title compound was prepared from commercially available **8** to afford after purification the desired product as a beige solid (34%). CAS # 1842362-42-8. ^1^H NMR (400 MHz, DMSO-*d_6_*) δ 8.53 (d, 1H, *J* = 6.8 Hz, H_6_), 7.55 (d, 1H, *J* = 9.3 Hz, H_9_), 7.33–7.25 (t, 1H, *J =* 9.3 Hz, H_8_), 7.17 (d, 2H, *J* = 8.7 Hz, 2 × H_arom_), 7.06–6.96 (m, 3H, H_7_, 2 × H_arom_), 3.81 (s, 3H, O–CH_3_) ppm.

3-(3-Fluorophenyl)pyrido [1,2-*e*]purine-2,4(*1H,3H*)-dione (**9g**). The title compound was prepared from commercially available **8** to afford after purification the desired product as a yellow solid (19%). ^1^H NMR (400 MHz, DMSO-*d*_6_) δ 8.50 (d, 1H, *J* = 6.2 Hz, H_6_), 7.55–7.49 (m, 2H, H_9_, H_arom_), 7.33–7.24 (m, 3H, H_8_, 2 × H_arom_), 7.20 (d, 1H, *J* = 9.7 Hz, H_arom_), 7.14 (d, 1H, *J* = 7.3 Hz, H_arom_), 6.97 (t, 1H, *J* = 6.2 Hz, H_7_) ppm. ^13^C NMR (101 MHz, DMSO-*d*_6_) δ 163.3 (C=O), 160.8 (C_quat_), 158.9 (d, *J =* 11.1 Hz, C_quat_), 151.7 (C=O), 141.6 (C_quat_), 130.0 (C_quat_), 129.9 (d, *J_C–F_* = 9.3 Hz, C_arom_), 126.2 (C_arom_), 125.7 (d, *J_C–F_ =* 3.03 Hz, C_arom_), 124.1 (C_arom_), 118.3 (C_arom_), 117.6 (C_quat_), 116.7 (d, *J_C–F_* = 22.7 Hz, C_arom_), 114.4 (d, *J_C–F_ =* 20.2 Hz, C_arom_), 112.4 (C_arom_) ppm. ^19^F NMR (376 MHz, DMSO-*d_6_*) δ −113.60 ppm. HRMS-ESI (*m/z*) [M+H]^+^ calcd. for C_15_H_9_FN_4_O_2_: 297.0782, found: 297.079. λ_abs_: 263 nm λ_em_: 498 nm λ_exc_: 266 nm. R*_f_* 0.29 (DCM/MeOH 9:1).

3-(3,4-Difluorophenyl)pyrido [1,2-*e*]purine-2,4(*1H,3H*)-dione (**9h**). The title compound was prepared from commercially available **8** to afford after purification the desired product as a white/yellow solid (38%). ^1^H NMR (400 MHz, DMSO-*d*_6_) δ 8.44 (d, 1H, *J =* 6.2 Hz, H_6_), 7.55–7.47 (m, 2H, 2 × H_arom_), 7.44 (dd, 1H, *J =* 7.4, 2.3 Hz, H_arom_), 7.25 (dd, 1H *J* = 9.4, 6.6 Hz, H_8_), 7.17–7.11 (m, 1H, H_arom_), 6.93 (t, 1H, *J =* 6.2 Hz, H_7_) ppm. ^13^C NMR (101 MHz, DMSO-*d*_6_) δ 159.2 (C=O), 152.5 (C=O), 150.1 (dd, *J* = 38.2, 12.9 Hz, C_quat_), 147.7 (dd, *J_C–F_* = 38.4, 12.8 Hz, C_quat_), 141.5 (C_quat_), 137.1 (C_quat_), 134.3 (dd, *J_C–F_* = 8.6, 3.5 Hz, C_quat_), 126.6 (dd, *J_C–F_* = 6.6, 3.4 Hz, C_arom_), 126.0 (C_arom_), 124.0 (C_arom_), 118.9 (d, *J_C–F_* = 18.2 Hz, C_arom_), 118.2 (C_arom_), 117.5 (C_quat_), 117.0 (d, *J_C–F_* = 17.9 Hz, C_arom_), 112.1 (C_arom_) ppm. ^19^F NMR (376 MHz, DMSO-*d_6_*) δ −140.24, −138.54 ppm. HRMS-ESI (*m/z*) [M+H]^+^ calcd. for C_15_H_9_F_2_N_4_O_2_: 315.0688, found: 315.0687. λ_abs_: 261 nm λ_em_: 49 nm λ_exc_: 266 nm. R*_f_* 0.27 (DCM/MeOH 9:1).

3-(2,4-Difluorophenyl)pyrido [1,2-*e*]purine-2,4(*1H,3H*)-dione (**9i**). The title compound was prepared from commercially available **8** to afford after purification the desired product as a yellow solid (64%).^1^H NMR (400 MHz, DMSO-*d*_6_) δ 8.45 (d, 1H, *J =* 6.9 Hz, H_6_), 7.58–7.36 (m, 3H, H_9_, 2 × H_arom_), 7.28 (t, 1H, *J =* 6.8 Hz, H_8_), 7.20 (t, 1H, *J =* 8.4 Hz, H_arom_), 6.95 (t, 1H, *J =* 6.9 Hz, H_7_) ppm. ^13^C NMR (101 MHz, DMSO-*d*_6_) δ 162.8 (dd, *J* = 46.5, 11.7 Hz, C=O), 158.5 (C_quat_), 159.6-156.3 (dd, *J =* 251.49, 13.2 Hz, C_quat_), 151.7 (C=O), 141.7 (C_quat_), 136.7 (C_quat_), 132.7 (dd, *J_C–F_* = 10.3, 2.4 Hz, C_arom_), 126.3 (C_arom_), 124.1 (C_arom_), 121.3 (dd, *J_C–F_* = 13.6, 4.0 Hz, C_quat_), 118.2 (C_arom_), 117.2 (C_quat_), 112.3 (C_arom_), 111.6 (dd, *J_C–F_* = 22.4, 3.6 Hz, C_arom_), 104.4 (dd, *J_C–F_* = 26.9, 24.6 Hz, C_arom_) ppm. ^19^F NMR (376 MHz, DMSO-*d_6_*) δ −110.21, −117.12 ppm. HRMS-ESI (*m/z*) [M+H]^+^ calcd. for C_15_H_9_F_2_N_4_O_2_: 315.0687, found: 315.0688. λ_abs_: 264 nm λ_em_: 490 nm λ_exc_: 266 nm. R*_f_* 0.32 (DCM/MeOH 9:1).

3-(3,4-Dimethylphenyl)pyrido [1,2-*e*]purine-2,4(*1H,3H*)-dione (**9j**). The title compound was prepared from commercially available 8 to afford after purification the desired product as a light-yellow solid (56%). ^1^H NMR (400 MHz, DMSO-*d_6_*) δ 8.46 (d, 1H, *J* = 6.9 Hz, H_6_), 7.54 (d, 1H, *J* = 9.3 Hz, H_9_), 7.28 (t, 1H, *J =* 8.0 Hz, H_arom_), 7.21 (d, 1H, *J* = 9.3 Hz, H_8_), 7.00–6.92 (m, 3H, H_7_, 2 × H_arom_), 2.27 (s, 3H, C–CH_3_). 2.24 (s, 3H, C–CH_3_) ppm. ^13^C NMR (101 MHz, DMSO-*d_6_*) δ 159.0 (C=O), 151.7 (C=O), 141.5 (C_quat_), 136.5 (C_quat_), 135.6 (C_quat_), 134.4 (C_quat_), 129.9 (C_arom_), 129.6 (C_arom_), 129.3 (C_arom_), 126.1 (C_arom_), 124.0 (C_arom_), 118.3 (C_arom_), 117.8 (C_quat_), 112.5 (C_arom_), 19.2 (C–CH_3_), 19.0 (C–CH_3_) ppm. HRMS-ESI (*m/z*) [M+H]^+^ calcd. for C_17_H_15_N_4_O_2_: 307.1189, found: 307.1190. λ_abs_: 263 nm λ_em_: 496 nm λ_exc_: 266 nm. R*_f_* 0.68 (DCM/MeOH, 9:1).

3-(4-Fluorophenyl)-2-thioxo-2,3-dihydropyrido [1,2-*e*]purin-4(*1H*)-one (**11**). The title compound was prepared from commercially available 8 to afford after purification the desired product as a light-yellow solid (74%). ^1^H NMR (400 MHz, DMSO-*d_6_*) δ 8.43 (d, 1H, *J* = 6.8 Hz, H_6_), 7.47 (d, 1H, *J* = 9.2 Hz, H_9_), 7.30–7.23 (m, 1H, H_8_), 7.20 (t, 2H, *J* = 8.5 Hz, 2 × H_arom_), 7.07–7.06 (m, 2H, 2 × H_arom_), 6.88 (t, 1H, *J* = 6.8 Hz, H_7_) ppm. ^13^C NMR (101 MHz, DMSO-*d*_6_) δ 175.6 (C=S), 161.9 (C=O), 159.5 (d, *J* = 10.9 Hz, C_quat_), 142.3 (C_quat_), 142.1 (C_quat_), 139.0 (C_quat_), 131.4 (d, *J_C–F_* = 8.6 Hz, 2 × C_arom_), 126.8 (C_arom_), 124.2 (C_arom_), 121.0(C_quat_), 118.0 (C_arom_), 115.1 (d, *J_C–F_* = 22.5 Hz, 2 × C_arom_), 111.4 (C_arom_) ppm. ^19^F NMR (376 MHz, DMSO-*d_6_*) δ −116.93 ppm. HRMS-ESI (*m/z*) [M+H]^+^ calcd. for C_15_H_10_FN_4_OS: 313.0553, found: 313.0553. λ_abs_: 340 nm λ_em_: 492 nm λ_exc_: 341 nm. R*_f_* 0.11 (DCM/MeOH 9:1).

3-(Benzyl)pyrido [1,2-*e*]purine-2,4(*1H,3H*)-dione (**10a**). The title compound was prepared from commercially available 8 to afford after purification the desired product as a light yellow solid (71%). CAS # 1842362-38-2. ^1^H NMR (400 MHz, DMSO-*d*_6_) δ 8.29 (d, 1H, *J* = 7.0 Hz, H_6_), 7.52 (dd, 1H, *J* = 8.3, 5.5 Hz, H_arom_), 7.41 (d, 1H, *J* = 9.3 Hz, H_9_), 7.35 (dd, 2H, *J* = 8.3, 5.5 Hz, 2 × H_arom_), 7.24 (t, 1H, *J* = 8.3 Hz, H_arom_), 7.16 (dd, 1H, *J* = 9.3, 6.75 Hz, H_8_), 7.08 (t, 2H, *J* = 8.8 Hz, 2× H_arom_), 6.81 (t, 1H, *J* = 6.7 Hz, H_7_), 5.09 (s, 2H, N–CH_2_) ppm.

3-(*p*-Fluorobenzyl)pyrido [1,2-*e*]purine-2,4(*1H,3H*)-dione (**10b**). The title compound was prepared from commercially available 8 to afford after purification the desired product as a light-yellow solid (52%). ^1^H NMR (400 MHz, DMSO-*d*_6_) δ 8.29 (d, 1H, *J* = 7.0 Hz, H_6_), 7.52 (dd, 1H, *J* = 8.3, 7.5 Hz, H_arom_), 7.41 (d, 1H, *J* = 9.3 Hz, H_9_), 7.35 (dd, 1H, *J =* 8.3, 5.7 Hz, H_arom_), 7.24 (t, 1H, *J* = 7.1 Hz, H_arom_), 7.21–7.12 (m, 1H, H_8_), 7.08 (t, 1H, *J* = 8.8 Hz, H_arom_), 6.81 (t, 1H, *J* = 7.0 Hz, H_7_), 5.09 (s, 2H, N–CH_2_) ppm. ^13^C NMR (101 MHz, DMSO-d_6_) δ 160.1 (C=O), 154.9 (C=O), 141.6 (C_quat_), 135.8 (C_quat_), 131.6 (d, *J_C–F_* = 8.4 Hz, C_arom_), 131.1 (C_quat_), 130.0 (d, *J_C–F_* = 8.1 Hz, 2 × C_arom_), 129.0 (C_quat_), 126.0 (C_arom_), 124.4 (C_arom_), 118.6 (C_arom_), 117.4 (C_quat_), 115.8 (d, *J_C–F_* = 21.1 Hz, C_arom_), 115.1 (d, *J_C–F_* = 21.5 Hz, C_arom_), 111.7 (C_arom_), 42.7 (d, *J_C–F_* = 91.0 Hz, N–CH_2_) ppm. ^19^F NMR (376 MHz, DMSO-*d_6_*) δ −116.00 ppm. HRMS-ESI (*m/z*) [M+H]^+^ calcd. for C_16_H_12_FN_4_O_2_: 311.0938, found: 311.0939. λ_abs_: 259 nm λ_em_: 501 nm λ_exc_: 268 nm. R*_f_* 0.37 (DCM/MeOH, 9:1).

3-(*p*-(Trifluoromethyl)benzyl)pyrido [1,2-*e*]purine-2,4(*1H,3H*)-dione (**10c**). The title compound was prepared from commercially available 8 to afford after purification the desired product as a white/yellow solid (46%). ^1^H NMR (400 MHz, DMSO-*d*_6_) δ 8.54 (d, 1H, *J* = 6.9 Hz, H_6_), 7.66 (d, 2H, *J* = 8.1 Hz, 2 × H_arom_), 7.53 (dd, 3H, *J* = 14.5, 8.8 Hz, H_9_, 2 × H_arom_), 7.31–7.26 (m, 1H, H_8_), 6.98 (t, 1H, *J* = 6.9 Hz, H_7_), 5.20 (s, 2H, N–CH_2_) ppm. ^13^C NMR (101 MHz, DMSO-*d*_6_) δ 158.6 (C=O), 150.9 (C=O), 142.7 (C_quat_), 141.7 (C_quat_), 127.9 (2 × C_arom_), 127.6 (C_quat_), 126.4 (C_quat_), 126.4 (C_arom_), 125.1 (t, *J_C–F_* = 3.7 Hz, 2 × C_arom_), 124.2 (C_arom_), 118.3 (C_arom_), 117.4 (m, C_quat_), 113.5 (C_quat_), 112.6 (C_arom_), 43.1 (N–CH_2_) ppm. ^19^F NMR (376 MHz, DMSO-*d_6_*) δ –60.72. HRMS-ESI (*m/z*) [M+H]^+^ calcd. for C_17_H_12_F_3_N_4_O_2_: 361.0906, found: 361.0906. λ_abs_: 260 nm λ_em_: 495 nm λ_exc_: 265 nm. R*_f_* 0.35 (DCM/MeOH, 9:1).

3-(*p*-Methylbenzyl)pyrido [1,2-*e*]purine-2,4(*1H,3H*)-dione (**10d**). The title compound was prepared from commercially available 8 to afford after purification the desired product as a light yellow solid (44%). ^1^H NMR (400 MHz, DMSO-*d*_6_) δ 8.49 (d, 1H, *J* = 7.0 Hz, H_6_), 7.54 (d, 1H, *J* = 9.5 Hz, H_9_), 7.28 (dd, 1H, *J* = 9.5, 6.5 Hz, H_8_), 7.21 (d, 2H, *J* = 7.9 Hz, 2 × H_arom_), 7.15–7.04 (m, 2H, 2 × H_arom_), 6.97 (dd, 1H, *J* = 7.0, 6.5 Hz, H_7_), 5.07 (s, 2H, N–CH_2_), 2.25 (s, 3H, C-CH_3_) ppm. ^13^C NMR (101 MHz, DMSO-*d*_6_) δ 161.7 (C=O), 158.7 (C_quat_), 154.5 (C=O), 141.6 (C_quat_), 135.9 (C_quat_), 135.0 (C_quat_), 128.6 (2 × C_arom_), 127.4 (2 × C_arom_), 126.2 (C_arom_), 124.2 (C_arom_), 118.2 (C_arom_), 117.6 (C_quat_), 112.5 (C_arom_), 43.0 (N–CH_2_), 20.6 (C–CH_3_) ppm. HRMS-ESI (*m/z*) [M+H]^+^ calcd. for C_17_H_15_N_4_O_2_: 307.1189, found: 307.1191. λ_abs_: 260 nm λ_em_: 496 nm λ_exc_: 265 nm. R*_f_* 0.46 (DCM/MeOH, 9:1).

3-Benzyl-2-thioxo-2,3-dihydropyrido [1,2-*e*]purin-4(*1H*)-one (**12**). The title compound was prepared from commercially available 8 to afford after purification the desired product as a light yellow solid (71%). CAS # 1842362-47-3. ^1^H NMR (250 MHz, DMSO-*d*_6_) δ 8.69 (d, 1H, *J* = 6.8 Hz, H_6_), 7.58 (d, 1H, *J* = 9.3 Hz, H_9_), 7.41–7.15 (m, 6H, H_8_, 5 × H_arom_), 7.01 (t, 1H, *J* = 6.8 Hz, H_7_), 5.79 (s, 2H, N–CH_2_) ppm.

3-(*p*-Fluorophenyl)-7-methylpyrido [1,2-*e*]purine-2,4(*1H,3H*)-dione (**18a**). The title compound was prepared from commercially available 16a to afford after purification the desired product as a yellow solid (66%). ^1^H NMR (400 MHz, DMSO-*d_6_*) δ 8.17 (s, 1H, H_6_), 7.41 (d, 1H, *J* = 9.4, Hz, H_9_), 7.26 (d, 4H, *J* = 6.5, Hz, 4 × H_arom_), 7.11 (d, 1H, *J* = 9.4 Hz, H_8_), 2.28 (s, 3H, C–CH_3_) ppm. ^13^C NMR (101 MHz, DMSO-*d*_6_) δ 162.8 (C=O), 159.2 (C_quat_), 152.8 (C_quat_), 152.9 (C=O), 141.8 (C_quat_), 136.5 (C_quat_), 133.7 (C_quat_), 131.2 (d, *J_C–F_* = 18.6 Hz, 2 × C_arom_), 123.2 (C_arom_), 117.3 (C_quat_), 115.8 (C_arom_), 115.6 (C_arom_), 114.7 (d, *J_C–F_* = 29.4 Hz, 2 × C_arom_), 21.0 (C–CH_3_) ppm. ^19^F NMR (376 MHz, DMSO-*d_6_*) δ −115.61 ppm. HRMS-ESI (*m/z*) [M+H]^+^ calcd for C_16_H_12_FN_4_O_2_: 311.0937, found: 311.0938. λ_abs_: 260 nm λ_em_: 497 nm λ_exc_: 265 nm. R*_f_* 0.28 (DCM/MeOH 9:1).

3-(*p*-Fluorophenyl)-8-methylpyrido [1,2-*e*]purine-2,4(*1H,3H*)-dione (**18b**). The title compound was prepared from commercially available 16b to afford after purification the desired product as a yellow solid (69%). ^1^H NMR (250 MHz, DMSO-*d_6_*) δ 8.31 (d, 1H, *J* = 7.0 Hz, H_6_), 7.31–7.24 (m, 5H, H_9_, 4 × H_arom_), 6.81 (d, 1H, *J* = 7.0 Hz, H_7_), 2.35 (s, 3H, C-CH_3_) ppm. ^13^C NMR (63 MHz, DMSO-*d*_6_) δ 162.4 (C=O), 159.3 (d, *J =* 121.2 Hz, C_quat_), 151.4 (C=O), 140.9 (C_quat_), 133.7 (C_quat_), 133.0 (d, *J* = 3.0 Hz, C_quat_), 131.3 (d, *J_C–F_* = 8.7 Hz, 2 × C_arom_), 129.5 (C_arom_), 121.8 (C_quat_), 120.8 (C_arom_), 117.8 (C_quat_), 117.6 (C_arom_), 115.5 (d, *J_C-F_* = 22.6 Hz, 2 × C_arom_), 17.8 (C–CH_3_) ppm. ^19^F NMR (376 MHz, DMSO-*d_6_*) δ −115.29 ppm. HRMS-ESI (*m/z*) [M+H]^+^ calcd for C_16_H_12_FN_4_O_2_: 311.0938, found: 311.0938. λ_abs_: 263 nm λ_em_: 465 nm λ_exc_: 269 nm. R*_f_* 0.28 (DCM/MeOH 9:1).

7-Bromo-3-(*p-*fluorophenyl)pyrido [1,2-*e*]purine-2,4(*1H,3H*)-dione (**19a**). The title compound was prepared from commercially available 17a to afford after purification the desired product as a yellow/orange solid (46%). ^1^H NMR (400 MHz, DMSO-*d*_6_) δ 8.36 (d, 1H, *J =* 7.4 Hz, H_9_), 7.85–7.77 (m, 1H, H_6_), 7.34–7.20 (m, 4H, 4 × H_arom_), 7.05 (dd, 1H, *J =* 7.4, 1.9 Hz*,* H_8_) ppm. ^13^C NMR (101 MHz, DMSO-*d*_6_) δ 163.0 (C=O), 159.4 (d, *J =* 17.2 Hz, C_quat_), 153.7 (C=O), 141.0 (C_quat_), 139.1 (C_quat_), 134.0 (d, *J_C–F_ =* 3.1 Hz, C_quat_), 131.2 (d, *J_C–F_* = 8.7 Hz, 2 × C_arom_), 125.0 (C_arom_), 120.0 (C_arom_), 118.8 (C_quat_), 117.8 (C_quat_), 115.3 (d, *J_C–F_* = 22.6 Hz, 2 × C_arom_), 115.2 (C_arom_) ppm. ^19^F NMR (376 MHz, DMSO-*d_6_*) δ −115.56 ppm. HRMS-ESI (*m/z*) [M+H]^+^ calcd for C_15_H_9_BrFN_4_O_2_ 374.9887, found: 374.9887. λ_abs_: 267 nm λ_em_: 492 nm λ_exc_: 269 nm. R*_f_* 0.32 (DCM/MeOH 9:1).

8-Bromo-3-(*p*-fluorophenyl)pyrido [1,2-*e*]purine-2,4(*1H,3H*)-dione (**19b**). The title compound was prepared from commercially available 17b to afford after purification the desired product as a yellow/green light solid (51%). ^1^H NMR (400 MHz, DMSO-*d_6_*) δ 8.74 (s, 1H, H_9_), 7.49 (d, 1H, *J* = 9.7 Hz, H_6_), 7.32 (d, 1H*, J* = 9.7 Hz, H_7_), 7.30–7.21 (m, 4H, 4 × H_arom_) ppm. ^13^C NMR (101 MHz, DMSO-*d*_6_) δ 162.8 (C=O), 159.7 (d, *J =* 70.7 Hz, C_quat_), 153.5 (C=O), 140.1 (C_quat_), 138.0 (C_quat_), 134.1 (d, *J_C–F_ =* 2.0 Hz, C_quat_), 131.7 (d, *J_C–F_ =* 8.8 Hz, 2 × C_arom_), 129.1 (C_arom_), 124.6 (C_arom_), 119.9 (C_arom_), 118.6 (C_quat_), 115.9 (d, *J_C–F_ =* 22.6 Hz, 2 × C_arom_), 106.3 (C_quat_) ppm. ^19^F NMR (376 MHz, DMSO-*d_6_*) δ −115.29 ppm. HRMS-ESI (*m/z*) [M+Na]^+^ calcd for C_15_H_9_BrFNaN_4_O_2_ 396.9706, found: 396.9693. λ_abs_: 262 nm λ_em_: 496 nm λ_exc_: 267 nm. R*_f_* 0.30 (DCM/MeOH 9:1).

### 3.4. General Synthetic Procedure 3 for the N^1^-Alkylation

To a solution of fluorobenzyl compound (150 mg) in anhydrous DMF (4 mL), were added subsequently potassium carbonate (1.5 eq.) and bromide derivative (1.5 eq.) under inert atmosphere. The reaction mixture was heated at 120 °C under microwave irradiation for 20 min. The mixture was dissolved with EtOAc, washed twice with saturated NH_4_Cl, dried over MgSO_4_, filtrated and concentrated under vacuum. Pure compound was obtained after purification by flash column chromatography using an elution gradient of DCM/MeOH (from 98:2 to 95:5).

1-Benzyl-3-(4-fluorophenyl)purino [9,8-*a*]pyridine-2,4-dione (**13a**). The title compound was prepared from commercially available 9a to afford after purification the desired product as a light orange solid (36%). ^1^H NMR (400 MHz, DMSO-*d_6_*) δ 8.23 (d, 1H, *J* = 7.0 Hz, H_6_), 7.60 (d, 1H, *J* = 9.3 Hz, H_9_), 7.50–7.27 (m, 9H, 5 × H_arom-N1_, 4 × H_arom-N3_), 7.26–7.20 (m, 1H, H_8_), 6.82 (t, 1H, *J* = 7.0 Hz, H_7_), 5.73 (s, 2H, N–CH_2_) ppm. ^13^C NMR (101 MHz, DMSO-*d_6_*) δ 160.3 (C=O), 157.9 (C_quat_), 151.1 (C=O), 142.5 (C_quat_), 136.1 (C_quat_), 132.6 (d, *J_C–F_* = 3.03 Hz, C_quat_), 132.08 (C_quat_), 131.1 (d, *J_C–F_* = 8.8 Hz, 2 × C_arom-N3_), 129.0 (2 × C_arom-N1_), 127.6 (C_arom_), 126.0 (C_arom_), 125.7 (2 × C_arom-N1_), 125.0 (C_arom_), 120.1 (C_quat_), 118.9 (C_arom_), 115.7 (d, *J_C–F_* = 22.8 Hz, 2 × C_arom-N3_), 113.4 (C_arom_), 46.9 (N–CH_2_) ppm. ^19^F NMR (376 MHz, DMSO-*d_6_*) δ −114.07 ppm. HRMS-ESI (*m/z*) [M+H]^+^ calcd. for C_22_H_16_FN_4_O_4_: 387.1251, found: 387.1249. λ_abs_: 261 nm λ_em_: 418 nm λ_exc_: 267 nm. R*_f_* 0.65 (DCM/MeOH 95:5).

3-(*p*-Fluorophenyl)-1-[(*p*-methylphenyl)methyl]purino[9,8-*a*]pyridine-2,4-dione (**13b**). The title compound was prepared from commercially available 9a to afford after purification the desired product as a light yellow solid (20%). ^1^H NMR (250 MHz, DMSO-*d_6_*) δ 8.20 (d, 1H, *J* = 7.3 Hz, H_6_), 7.56 (d, 1H, *J* = 9.3 Hz, H_9_), 7.41 (dd, 2H, *J* = 9.0, 5.2 Hz, 2 × H_arom-N3_), 7.38–7.25 (m, 4H, 2 × H_arom-N1_, 2 × H_arom-N3_), 7.24–7.09 (m, 3H, H_8_, 2 × H_arom-N1_), 6.79 (t, 1H, *J* = 7.3 Hz, H_7_), 5.64 (s, 2H, N–CH_2_), 2.23 (s, 3H, C–CH_3_) ppm. ^13^C NMR (63 MHz, DMSO-*d_6_*) δ 159.5 (C=O), 157.9 (C_quat_), 151.1 (C=O), 142.5 (C_quat_), 136.7 (C_quat_), 133.0 (C_quat_), 132.7 (d, *J_C–F_ =* 3.1 Hz, C_quat_), 132.0 (C_quat_), 131.1 (d, *J_C–F_ =* 8.8 Hz, 2 × C_arom-N3_), 129.5 (2 × C_arom-N1_), 126.0 (C_arom_), 125.5 (2 × C_arom-N1_), 125.1 (C_arom_), 120.1 (C_quat_), 118.9 (C_arom_), 115.7 (d, *J_C–F_ =* 23.3 Hz, 2 × C_arom-N3_), 113.4 (C_arom_), 46.7 (N–CH_2_), 20.6 (C–CH_3_) ppm. ^19^F NMR (376 MHz, DMSO-*d_6_*) δ −114.08 ppm. HRMS-ESI (*m/z*) [M+H]^+^ calcd. for C_23_H_18_FN_4_O_2_: 401.1708, found: 401.1405. λ_abs_: 260 nm λ_em_: 418 nm λ_exc_: 261 nm. R*_f_* 0.5 (DCM/MeOH 95:5).

4-[[3-(*p*-Fluorophenyl)-2,4-dioxo-purino[9,8-*a*]pyridin-1-yl]methyl]benzonitrile (**13c**). The title compound was prepared from commercially available 9a to afford after purification the desired product as a light yellow solid (30%). ^1^H NMR (400 MHz, DMSO-*d_6_*) δ 8.13 (d, 1H, *J* = 7.3 Hz, H_6_), 7.86 (d, 2H, *J* = 8.1 Hz, 2 × H_arom-N1_), 7.69 (d, 2H, *J* = 8.1 Hz, 2 × H_arom-N1_), 7.62 (d, 1H, *J* = 9.3 Hz, H_9_), 7.45–7.43 (m, 2H, 2 × H_arom-N3_), 7.34 (t, 2H, *J* = 8.7 Hz, 2 × H_arom-N3_), 7.29–7.21 (m, 1H, H_8_), 6.83 (t, 1H, *J* = 7.3 Hz, H_7_), 5.80 (s, 2H, N–CH_2_) ppm. ^13^C NMR (101 MHz, DMSO-*d_6_*) δ 160.3 (C=O), 157.9 (C_quat_), 151.1 (C=O), 142.5 (C_quat_), 142.0 (C_quat_), 132.8 (2 × C_arom-N1_), 132.6 (d, *J_C–F_* = 3.0 Hz, C_quat_), 131.8 (C_quat_), 131.1 (d, *J_C–F_* = 8.8 Hz, 2 × C_arom-N3_), 126.9 (2 × C_arom-N1_), 126.1 (C_arom_), 124.7 (C_arom_), 120.3 (C_quat_), 119.0 (C_arom_), 118.5 (C_quat_), 115.7 (d, *J_C–F_* = 22.8 Hz, 2 × C_arom-N3_), 113.7 (C_arom_), 110.4 (C_quat_), 47.0 (N–CH_2_) ppm. ^19^F NMR (376 MHz, DMSO-*d_6_*) δ −114.00 ppm. HRMS-ESI (*m/z*) [M+H]^+^ calcd. for C_23_H_15_FN_5_O_2_: 412.1203, found: 412.1204. λ_abs_: 259 nm λ_em_: 418 nm λ_exc_: 259 nm. R*_f_* 0.5 (DCM/MeOH 95:5).

3-(*p*-Fluorophenyl)-1-[(*p*-nitrophenyl)methyl]purino [9,8-*a*]pyridine-2,4-dione (**13d**). The title compound was prepared from commercially available **9a** to afford after purification the desired product as a yellow solid (28%). ^1^H NMR (250 MHz, DMSO-*d_6_*) δ 8.19 (d, 2H, *J* = 8.8 Hz, 2 × H_arom-N1_), 7.95 (d, 1H, *J* = 7.3 Hz, H_6_), 7.77 (d, 2H, *J* = 8.8 Hz, 2 × H_arom-N1_), 7.63 (d, 1H, *J* = 9.3 Hz, H_9_), 7.50–7.20 (m, 5H, H_8_, 4 × H_arom-N3_), 6.84 (t, 1H, *J* = 7.3 Hz, H_7_), 5.85 (s, 2H, N-CH_2_) ppm. ^13^C NMR (63 MHz, DMSO-*d_6_*) δ 159.5 (C=O), 157.9 (C_quat_), 151.1 (C=O), 147.0 (C_quat_), 144.0 (C_quat_), 142.5 (C_quat_), 132.5 (d, *J_C–F_ =* 3.1 Hz, C_quat_), 131.8 (C_quat_), 131.1 (d, *J_C–F_ =* 8.8 Hz, 2 × C_arom-N3_), 127.2 (2 × C_arom-N1_), 126.1 (C_arom_), 124.7 (C_arom_), 123.9 (2 × C_arom-N1_), 120.3 (C_quat_), 119.06 (C_arom_), 115.7 (d, *J_C–F_ =* 23.3 Hz, 2 × C_arom-N3_), 113.7 (C_arom_), 46.9 (N–CH_2_) ppm. ^19^F NMR (376 MHz, DMSO-*d_6_*) δ −113.96 ppm. HRMS-ESI (*m/z*) [M+H]^+^ calcd. for C_22_H_15_FN_5_O_4_: 432.1002, found: 432.1100. λ_abs_: 269.5 nm λ_em_: 551 nm λ_exc_: 275 nm. R*_f_* 0.5 (DCM/MeOH 95:5).

Methyl 4-[[3-(*p*-fluorophenyl)-2,4-dioxo-purino [9,8-*a*]pyridin-1-l]methyl]benzoate (**13e**). The title compound was prepared from commercially available **9a** to afford after purification the desired product as a white solid (65%). ^1^H NMR (400 MHz, DMSO-*d_6_*) δ 8.14 (d, 1H, *J* = 7.3 Hz, H_6_), 7.95 (d, 2H, *J* = 7.9 Hz, 2 × H_arom-N1_), 7.61 (d, 3H, *J* = 7.9 Hz, H_9_, 2 × H_arom-N1_), 7.45 (dd, 2H, *J* = 8.6, 5.3 Hz, 2 × H_arom-N3_), 7.35 (t, 2H, *J* = 8.6 Hz, 2 × H_arom-N3_), 7.28–7.20 (m, 1H, H_8_), 6.82 (t, 1H, *J* = 7.3 Hz, H_7_), 5.80 (s, 2H, N–CH_2_), 3.83 (s, 3H, O–CH_3_) ppm. ^13^C NMR (101 MHz, DMSO-*d_6_*) δ 165.8 (C=O), 157.9 (C=O), 151.1 (C=O), 142.5 (C_quat_), 141.8 (C_quat_), 132.6 (d, *J_C–F_* = 3.2 Hz, C_quat_), 131.9 (C_quat_), 131.1 (d, *J_C–F_* = 8.9 Hz, 2 × C_arom-N3_), 129.7 (2 × C_arom-N1_), 128.9 (C_quat_), 126.2 (2 × C_arom-N1_), 126.1 (C_arom_), 125.0 (C_quat_), 124.8 (C_arom_), 120.2 (C_quat_), 119.0 (C_arom_), 115.7 (d, *J_C–F_* = 22.7 Hz, 2 × C_arom-N3_), 113.6 (C_arom_), 52.1 (O–CH_2_), 47.01 (N–CH_3_) ppm. ^19^F NMR (376 MHz, DMSO-*d_6_*) δ −114.03 ppm. HRMS-ESI (*m/z*) [M+H]^+^ calcd. for C_24_H_18_FN_4_O_4_: 445.1306 found: 445.1305. λ_abs_: 259 nm λ_em_: 492 nm λ_exc_: 266 nm. R*_f_* 0.45 (DCM/MeOH 95:5).

1-[[*p*-(Dimethoxyphosphorylmethyl)phenyl]methyl]-3-(4-fluorophenyl)purino [9,8-*a*]pyridine-2,4-dione (**13f**). The title compound was prepared from commercially available **9a** to afford after purification the desired product as a yellow foam (16%). ^1^H NMR (400 MHz, DMSO-*d_6_*) δ 8.22 (d, 1H, *J* = 7.3 Hz, H_6_), 7.60 (d, 1H, *J* = 9.3 Hz, H_9_), 7.46 (dd, 2H, *J* = 7.8, 5.3 Hz, 2 × H_arom-N3_), 7.41–7.31 (m, 4H, 2 × H_arom-N1_, 2 × H_arom-N3_), 7.31–7.20 (m, 3H, H_8_, 2 × H_arom-N1_), 6.81 (t, 1H, *J* = 7.3 Hz, H_7_), 5.71 (s, 2H, N–CH_2_), 3.55 (s, 3H, O–CH_3_), 3.53 (s, 3H, O–CH_3_), 3.33 (s, 2H, C–CH_2_) ppm. ^13^C NMR (101 MHz, DMSO-d_6_*)* δ 160.8 (C=O), 158.4 (C_quat_), 151.6 (C=O), 143.0 (C_quat_), 134.9 (d, *J* = 3.7 Hz, C_quat_), 133.1 (d, *J_C–F_* = 3.0 Hz, C_quat_), 132.5 (C_quat_), 131.9 (d, *J_C–F_* = 9.1 Hz, C_quat_), 131.6 (d, *J_C–F_* = 9.0 Hz, 2 × C_arom-N3_), 130.8 (d, *J_C–P_* = 6.5 Hz, 2 × C_arom-N1_), 126.5 (C_arom_), 126.2 (d, *J_C–P_* = 3.0 Hz, 2 × C_arom-N1_), 125.5 (C_arom_), 120.6 (C_quat_), 119.4 (C_arom_), 116.2 (d, *J_C–F_* = 22.9 Hz, 2 × C_arom-N3_), 113.9 (C_arom_), 52.8 (O–CH_3_), 52.7 (O–CH_3_), 47.2 (N–CH_2_), 29.52 (d, *J^1^_C–P_ =* 108.5 Hz, P–CH_2_) ppm. ^19^F NMR (376 MHz, DMSO-*d_6_*) δ -114.07 ppm. ^31^P NMR (162 MHz, DMSO-*d_6_*) δ 28.89 ppm. HRMS-ESI (*m/z*) [M+H]^+^ calcd. for C_24_H_21_FN_4_O_5_P: 509.1384, found: 509.1380.λ_abs_: 264 nm λ_em_: 415 nm λ_exc_: 265 nm. R*_f_* 0.2 (DCM/MeOH 95:5).

### 3.5. General Synthetic Procedure 4 for the Sonogashira Coupling cross Coupling

Under inert atmosphere, to a solution of deprotected Br-compound (1 eq.) in dry DMF (0.082 M) were successively added copper iodide (0.2 eq.), triethylamine (3 eq.), alkynyl substrate (3 eq.), and Pd(PPh_3_)_4_ (10 mol%). The reaction mixture was heated under microwave irradiation at 110 °C for 15 min. The reaction was quenched with EtOAc and co-evaporated with heptane. Pure compounds were obtained after purification by flash column chromatography with DCM/MeOH as eluent.

3-(*p*-Fluorophenyl)-7-(4-phenylbut-1-ynyl)-*1H*-purino [9,8-*a*]pyridine-2,4-dione (**20a**). The title compound was prepared from commercially available 19a to afford after purification the desired product as an orange/yellow solid (70%). ^1^H NMR (400 MHz, DMSO-*d_6_*) δ 8.35 (d, 1H, *J* = 7.2 Hz, H_9_), 7.45 (s, 1H, H_6_), 7.35–7.30 (d, 4H, 4 × H_arom_), 7.29–7.22 (m, 5H, 4 × H_arom-N3_, H_arom_), 6.77 (dd, 1H, *J* = 7.2, 1.8 Hz, H_8_), 2.88 (t, 2H, *J* = 7.2 Hz, C–CH_2_), 2.76 (t, 2H, *J* = 7.2 Hz, CH_2_–CH_2_) ppm. ^13^C NMR (101 MHz, DMSO-*d_6_*) δ 162.3 (C=O), 159.9 (C_quat_), 159.2 (C_quat_), 153.2 (C=O), 140.7 (C_quat_), 140.3 (C_quat_), 134.0–133.6 (m, C_quat_), 131.2 (d, *J_C–F_* = 8.8 Hz, 2 × C_arom-N3_), 128.5 (2 × C_arom_), 128.2 (2 × C_arom_), 126.2 (C_arom_), 123.9 (C_arom_), 120.4 (C_quat_), 119.9 (C_arom_), 118.6 (C_quat_), 115.3 (d, *J_C–F_* = 22.7 Hz, 2 × C_arom-N3_), 114.0 (C_arom_), 93.9 (CH_2_–C), 80.0 (C≡C), 34.0 (CH_2_–C), 21.0 (C–CH_2_) ppm. ^19^F NMR (376 MHz, DMSO-*d_6_*) δ −115.39 ppm. HRMS-ESI (*m/z*) [M+H]^+^ calcd. for C_25_H_18_FN_4_O_2_: 425.1408, found: 425.1409. λ_abs_: 266 nm λ_em_: 513 nm λ_exc_: 278 nm. R*_f_* 0.34 (DCM/MeOH 95:5).

3-(*p*-Fluorophenyl)-7-[2-(4-propylphenyl)ethynyl]-*1H*-purino [9,8-*a*]pyridine-2,4-dione (**20b**). The title compound was prepared from commercially available **19a** to afford after purification the desired product as a yellow solid (74%). ^1^H NMR (400 MHz, DMSO-*d*_6_) δ 9.57 (s, 1H, H_1_), 8.39 (d, 1H, *J* = 7.3 Hz, H_9_), 7.71 (s, 1H, H_6_), 7.52 (d, 2H, *J* = 7.8 Hz, 2 × H_arom_), 7.28 (dd, 6H, *J* = 7.5, 3.5 Hz, 4 × H_arom-N3_, 2 × H_arom_), 6.99 (d, 1H, *J* = 7.3 Hz, H_8_), 2.60 (t, 2H, *J* = 7.5 Hz, C–CH_2_), 1.61 (h, 2H, *J* = 7.5 Hz, CH_2_–CH_2_), 0.90 (t, 3H, *J* = 7.5 Hz, CH_2_–CH_3_) ppm. ^13^C NMR (101 MHz, DMSO-*d*_6_) δ 162.3 (C=O), 159.9 (C_quat_), 159.2 (C_quat_), 154.3 (C=O), 143.6 (C_quat_), 140.6 (C_quat_), 133.6 (C_quat_), 131.4 (2 × C_arom_), 131.2 (d, *J_C–F_* = 8.7 Hz, 2 × C_arom-N3_), 128.8 (2 × C_arom_), 123.9 (C_arom_), 120.9 (C_arom_), 119.0 (C_quat_), 118.9 (C_quat_), 115.3 (d, *J_C–F_* = 22.3 Hz, 2 × C_arom-N3_), 113.8 (C_arom_), 92.6 (C_quat_), 90.1 (CH_2_–C), 87.6 (C≡C), 37.0 (C–CH_2_), 23.7 (CH_2_–CH_2_), 13.5 (CH_2_–CH_3_) ppm. ^19^F NMR (376 MHz, DMSO-*d_6_*) δ −115.43 ppm. HRMS-ESI (*m/z*) [M+H]^+^ calcd. for C_26_H_20_FN_4_O_2_: 439.1564, found: 439.1563. λ_abs_: 300 nm λ_em_: 528 nm λ_exc_: 307 nm. R*_f_* 0.35 (DCM/MeOH 95:5).

3-(*p*-Fluorophenyl)-8-hex-1-ynyl-*1H*-purino [9,8-*a*]pyridine-2,4-dione (**20c**). The title compound was prepared from commercially available 19a to afford after purification the desired product as a yellow/orange solid (91%). ^1^H NMR (400 MHz, DMSO-*d_6_*) δ 12.95 (bs, 1H, H_1_), 8.40 (d, 1H, *J* = 7.2 Hz, H_9_), 7.61 (s, 1H, H_6_), 7.39–7.25 (m, 4H, 4 × H_arom_), 6.95 (dd, 1H, *J* = 7.2, 1.6 Hz, H_8_), 2.51–2.47 (m, 2H, CH_2_), 1.59–1.52 (m, 2H, CH_2_), 1.50–1.39 (m, 2H, CH_2_), 0.93 (t, 3H, *J* = 7.3 Hz, CH_2_–CH_3_) ppm. ^13^C NMR (101 MHz, DMSO-*d_6_*) δ 161.6 (C=O), 159.5 (C_quat_), 150.0 (C=O), 141.1 (C_quat_), 132.4 (C_quat_), 131.3 (d, *J_C–F_* = 9.0 Hz, 2 × C_arom_), 123.8 (C_arom_), 120.7 (C_quat_), 120.5 (C_arom_), 119.9 (C_quat_), 115.6 (d, *J_C–F_* = 23.2 Hz, 2 × C_arom_), 115.3 (C_arom_), 113.7 (C_quat_), 95.0 (CH_2_–C), 79.1 (C≡C), 30.0 (CH_2_–CH_2_), 21.4 (CH_2_–CH_2_), 18.4 (CH_2_–CH_2_), 13.4 (CH_2_–CH_3_) ppm. ^19^F NMR (376 MHz, DMSO-*d_6_*) δ −114.41 ppm. HRMS-ESI (*m/z*) [M+H]^+^ calcd. for C_21_H_18_FN_4_O_2_: 377.1408, found: 377.1406. λ_abs_: 263 nm λ_em_: 513nm λ_exc_: 278 nm. R*_f_* 0.33 (DCM/MeOH 95:5).

3-(*p*-Fluorophenyl)-8-hept-1-ynyl-*1H*-purino [9,8-*a*]pyridine-2,4-dione (**20d**). The title compound was prepared from commercially available 19a to afford after purification the desired product as an orange solid (95%). ^1^H NMR (250 MHz, DMSO-*d*_6_) δ 12.95 (s, 1H, H_1_), 8.43 (d, 1H, *J* = 7.3 Hz, H_9_), 7.62 (s, 1H, H_5_), 7.42–7.24 (m, 4H, 4 × H_arom_), 6.97 (dd, 1H, *J* = 7.3, 1.6 Hz, H_8_), 2.51–2.47 (m, 2H, CH_2_), 1.57 (q, 2H, *J* = 7.0 Hz, CH_2_–CH_2_), 1.46–1.28 (m, 4H, 2 × CH_2_), 0.90 (t, 3H, *J* = 7.0 Hz, CH_2_–CH_3_) ppm. ^13^C NMR (63 MHz, DMSO-*d*_6_) δ 163.4 (C=O), 158.2 (C_quat_), 150.2 (C=O), 141.1 (C_quat_), 132.3 (d, *J_C–F_* = 3.7 Hz, C_quat_), 131.3 (d, *J_C–F_* = 8.8 Hz, 2 × C_arom_), 128.6 (C_quat_), 123.8 (C_arom_), 120.8 (C_arom_), 120.5 (C_quat_), 119.0 (C_quat_), 115.9 (2 × C_arom_), 115.5 (C_arom_), 95.2 (CH_2_–C), 79.0 (C≡C), 30.4 (C–CH_2_), 27.5 (CH_2_–CH_2_), 21.5 (CH_2_–CH_2_), 18.8 (CH_2_–CH_2_), 13.8 (CH_2_–CH_3_) ppm. ^19^F NMR (376 MHz, DMSO-*d_6_*) δ −114.36 ppm. HRMS-ESI (*m/z*) [M+H]^+^ calcd. for C_22_H_20_FN_4_O_2_: 391.1564, found: 391.1563. λ_abs_: 267 nm λ_em_: 517 nm λ_exc_: 277 nm. R*_f_* 0.33 (DCM/MeOH 95:5).

3-(*p*-Fluorophenyl)-7-oct-1-ynyl-*1H*-purino [9,8-*a*]pyridine-2,4-dione (**20e**). The title compound was prepared from commercially available 19a to afford after purification the desired product as a yellow solid (92%). ^1^H NMR (400 MHz, DMSO-*d_6_*) δ 8.41 (d, 1H, *J* = 7.7 Hz, H_9_), 7.61 (s, 1H, H_6_), 7.39–7.27 (m, 4H, 4 × H_arom_), 6.96 (dd, 1H, *J* = 7.7, 1.5 Hz, H_8_), 2.51–2.40 (m, 2H, CH_2_), 1.57 (p, 2H, *J* = 7.0 Hz, CH_2_), 1.43 (dt, 2H, *J* = 13.3, 7.0 Hz, CH_2_–CH_2_), 1.30 (dt, 4H, *J* = 7.0, 3.8 Hz, 2 × CH_2_), 0.89 (t, 3H, *J* = 7.0 Hz, CH_2_–CH_3_) ppm. ^13^C NMR (101 MHz, DMSO-*d_6_*) δ 162.7 (C=O), 158.4 (C_quat_), 150.3 (C=O), 141.2 (C_quat_), 132.4 (C_quat_), 132.3 (d, *J_C–F_* = 3.1 Hz, C_quat_), 131.2 (d, *J_C–F_* = 8.8 Hz, 2 × C_arom_), 123.8 (C_arom_), 120.8 (C_quat_), 120.5 (C_arom_), 118.9 (C_quat_), 115.6 (d, *J_C–F_* = 22.7 Hz, 2 × C_arom_), 115.4 (C_arom_), 95.1 (CH_2_–C), 79.1 (C≡C), 30.7 (C–CH_2_), 27.9 (CH_2_–CH_2_), 27.8 (CH_2_–CH_2_), 21.9 (CH_2_–CH_2_), 18.7 (CH_2_–CH_2_), 13.9 (CH_2_–CH_3_) ppm. ^19^F NMR (376 MHz, DMSO-*d_6_*) δ −114.38 ppm. HRMS-ESI (*m/z*) [M+H]^+^ calcd. for C_23_H_22_FN_4_O_2_: 405.1721, found: 405.1718. λ_abs_: 268 nm λ_em_: 512 nm λ_exc_: 276 nm. R*_f_* 0.33 (DCM/MeOH 95:5).

*N*-[3-[3-(*p*-fluorophenyl)-2,4-dioxo-*1H*-purino [9,8-*a*]pyridin-7-yl]prop-2-ynyl]octanamide (**20f**). The title compound was prepared from commercially available **19a** to afford after purification the desired product as an orange solid (95%). ^1^H NMR (250 MHz, DMSO-*d*_6_) δ 8.43–8.31 (m, 1H, H_9_), 7.52 (s, 1H, H_6_), 7.26 (t, 4H, *J* = 7.1 Hz, 4 × H_arom_), 6.82 (t, 1H, *J* = 8.0 Hz, H_8_), 4.23 (d, 2H, *J* = 11.3 Hz, C–CH_2_), 2.10 (q, 2H, *J* = 12.5, 9.9 Hz, CO–CH_2_), 1.65 (q, 2H, *J* = 7.1 Hz, CH_2_–CH_2_), 1.24 (q, 8H, *J* = 9.6, 7.1 Hz, 4 × CH_2_), 0.85 (tq, 3H, *J* = 11.1, 6.5, 5.6 Hz, CH_2_–CH_3_) ppm. ^13^C NMR (63 MHz, DMSO-*d*_6_) δ 167.0 (C=O), 162.7 (C=O), 159.3 (C_quat_), 152.03 (C=O), 136.1 (C_quat_), 134.0 (C_quat_), 133.8 (d, *J_C–F_* = 3.2 Hz, C_quat_), 131.3 (d, *J_C–F_* = 11.6 Hz, 2 × C_arom_), 124.1 (C_arom_), 120.9 (C_arom_), 119.0 (C_quat_), 118.8 (C_quat_), 115.2 (2 × C_arom_), 113.7 (C_arom_), 90.9 (CH_2_–C), 79.6 (C≡C), 45.3 (N–CH_2_), 34.6 (CO–CH_2_), 31.0 (CH_2_–CH_2_), 28.5 (CH_2_–CH_2_), 28.2 (CH_2_–CH_2_), 24.9 (CH_2_–CH_2_), 22.1 (CH_2_-CH_2_), 13.7 (CH_2_–CH_3_) ppm. ^19^F NMR (376 MHz, DMSO-*d_6_*) δ −115.58 ppm. HRMS-ESI (*m/z*) [M+H]^+^ calcd. for C_26_H_27_FN_5_O_2_: 476.2092, found: 476.2089. λ_abs_: 271 nm λ_em_: 510 nm λ_exc_: 279 nm. R*_f_* 0.32 (DCM/MeOH 95:5).

1-[3-[3-(*p*-Fluorophenyl)-2,4-dioxo-*1H*-purino [9,8-*a*]pyridin-7-yl]prop-2-ynyl]-3-hexyl-urea (**20g**). The title compound was prepared from commercially available 19a to afford after purification the desired product as an orange solid (92%). ^1^H NMR (250 MHz, DMSO-*d_6_*) δ 8.29 (d, 1H, *J =* 7.5 Hz, H_9_), 7.48 (s, 1H, H_6_), 7.41–7.11 (m, 4H, 4 × H_arom-N3_), 6.73 (t, 1H, *J* = 7.5 Hz, H_8_), 4.09 (d, 1H, *J* = 5.7 Hz, NH–CH_2_), 1.52–0.99 (m, 10H, 5 × CH_2_), 0.85 (t, 3H, *J* = 6.6 Hz, CH_2_–CH_3_) ppm. ^13^C NMR (63 MHz, DMSO-*d_6_*) δ 163.8 (C=O), 158.0 (C=O), 151.4 (C=O), 140.7 (C_quat_), 136.9 (C_quat_), 135.3 (C_quat_), 131.6 (d, *J_C–F_* = 9.9 Hz, 2 × C_arom_), 124.3 (C_arom_), 121.2 (C_arom_), 118.9 (C_quat_), 115.7 (C_quat_), 115.2 (2 × C_arom_), 113.3 (C_arom_), 96.3 (C_quat_), 92.1 (CH_2_–C), 80.6 (C≡C), 39.7 (N–CH_2_), 31.4 (CH_2_–CH_2_), 30.4 (CH_2_–CH_2_), 30.0 (CH_2_–CH_2_), 26.4 (CH_2_–CH_2_), 22.4 (CH_2_–CH_2_), 14.2 (CH_2_–CH_3_) ppm. ^19^F NMR (376 MHz, DMSO-*d_6_*) δ −116.27 ppm. HRMS-ESI (*m/z*) [M+H]^+^ calcd. for C_25_H_26_FN_6_O_3_: 477.2044, found: 477.2042. λ_abs_: 279 nm λ_em_: 515 nm λ_exc_: 278 nm. R*_f_* 0.30 (DCM/MeOH 95:5).

3-(*p*-Fluorophenyl)-8-(4-phenylbut-1-ynyl)-*1H*-purino [9,8-*a*]pyridine-2,4-dione (**21a**). The title compound was prepared from commercially available 19b to afford after purification the desired product as an orange solid (80%). ^1^H NMR (250 MHz, DMSO-*d_6_*) δ 8.44 (s, 1H, H_9_), 7.45 (d, 1H, *J* = 9.6 Hz, H_6_), 7.33 (d, 4H, *J* = 4.3 Hz, 4 × H_arom_), 7.25 (h, 5H, *J* = 4.0 Hz, 4 × H_arom-N3_, H_arom_), 7.07 (d, 1H, *J* = 9.6 Hz, H_7_), 2.89 (t, 2H, *J* = 6.9 Hz, C–CH_2_), 2.75 (t, 2H, *J* = 6.9 Hz, CH_2_–CH_2_) ppm. ^13^C NMR (63 MHz, DMSO-*d_6_*) δ 162.4 (C=O), 159.3 (C_quat_), 153.7 (C=O), 151.2 (C_quat_), 140.3 (C_quat_), 139.8 (C_quat_), 133.9 (C_quat_), 131.2 (2 × C_arom_), 128.5 (d, *J_C–F_* = 8.8 Hz, 2 × C_arom-N3_), 128.2 (2 × C_arom_), 127.9 (C_arom_), 126.3 (C_arom_), 126.2 (C_arom_), 118.3 (C_arom_), 118.0 (C_quat_), 115.3 (d, *J_C–F_* = 22.3 Hz, 2 × C_arom-N3_), 107.8 (C_quat_), 91.7 (CH_2_–C), 77.4 (C≡C), 34.0 (C–CH_2_), 20.8 (CH_2_–C) ppm. ^19^F NMR (376 MHz, DMSO-*d_6_*) δ −115.57 ppm. HRMS-ESI (*m/z*) [M+H]^+^ calcd. for C_25_H_18_FN_4_O_2_: 425.1408, found: 425.1406. λ_abs_: 272 nm λ_em_: 443 nm λ_exc_: 286 nm. R*_f_* 0.33 (DCM/MeOH 95:5).

3-(*p*-Fluorophenyl)-8-[2-(4-propylphenyl)ethynyl]-*1H*-purino [9,8-*a*]pyridine-2,4-dione (**21b**). The title compound was prepared from commercially available **19b** to afford after purification the desired product as a beige solid (67%). ^1^H NMR (250 MHz, DMSO-*d_6_*) δ 8.81 (s, 1H, H_9_), 7.86–7.72 (m, 1H, H_6_), 7.68–7.61 (m, 3H, H_7_, 2 × H_arom_), 7.55–7.48 (m, 2H, 2 × H_arom_), 7.44–7.23 (m, 4H, 4 × H_arom-N3_), 2.61 (t, 2H, *J* = 7.6 Hz, C–CH_2_), 1.59 (ddt, 2H, *J* = 12.8, 10.0, 6.7 Hz, CH_2_–CH_2_), 0.98–0.80 (m, 3H, CH_2_–CH_3_) ppm. ^13^C NMR (63 MHz, DMSO-*d_6_*) δ 163.4 (C=O), 159.6 (C_quat_), 150.0 (C=O), 143.6 (C_quat_), 140.4 (C_quat_), 133.7 (C_quat_), 132.2 (C_quat_), 131.4 (2 × C_arom_), 131.3 (d, *J_C–F_* = 10.3 Hz, 2 × C_arom-N3_), 128.8 (2 × C_arom_), 128.8 (C_arom_), 126.9 (C_arom_), 118.7 (C_arom_), 118.5 (C_quat_), 118.5 (C_quat_), 115.5 (d, *J_C–F_* = 24.9 Hz, 2 × C_arom-N3_), 108.5 (C_quat_), 91.2 (CH_2_–C), 84.7 (C≡C), 37.0 (CH_2_–CH_2_), 23.6 (CH_2_–CH_2_), 13.4 (CH_2_–CH_3_) ppm. ^19^F NMR (235 MHz, DMSO-*d_6_*) δ −114.29 ppm. HRMS-ESI (*m/z*) [M+H]^+^ calcd. for C_26_H_20_FN_4_O_2_: 439.1564, found: 439.1562. λ_abs_: 302 nm λ_em_: 454 nm λ_exc_: 288 nm. R*_f_* 0.35 (DCM/MeOH 95:5).

3-(*p*-Fluorophenyl)-8-hex-1-ynyl-*1H*-purino [9,8-*a*]pyridine-2,4-dione (**21c**). The title compound was prepared from commercially available 19b to afford after purification the desired product as a beige solid (42%). ^1^H NMR (400 MHz, DMSO-*d_6_*) δ 12.80 (s, 1H, H_1_), 8.64 (s, 1H, H_9_), 7.56 (d, 1H, *J* = 9.5 Hz, H_6_), 7.38–7.27 (m, 4H, 4 × H_arom_), 7.23 (dd, 1H, *J* = 9.5, 1.6 Hz, H_8_), 2.48 (d, 2H, *J* = 7.0 Hz, C–CH_2_), 1.61–1.51 (m, 2H, CH_2_–CH_2_), 1.51–1.41 (m, 2H, CH_2_–CH_2_), 0.94 (t, 3H, *J* = 7.0 Hz, CH_2_–CH_3_) ppm. ^13^C NMR (101 MHz, DMSO-*d_6_*) δ 160.3 (C=O), 158.4 (C_quat_), 150.2 (C=O), 140.3 (C_quat_), 132.3 (C_quat_), 132.2 (C_quat_), 131.2 (d, *J_C–F_* = 8.8 Hz, 2 × C_arom_), 128.7 (C_arom_), 126.5 (C_arom_), 118.3 (C_quat_), 118.3 (C_arom_), 115.6 (d, *J_C–F_* = 22.8 Hz, 2 × C_arom_), 109.2 (C_quat_), 92.8 (CH_2_–C), 76.4 (C≡C), 30.0 (CH_2_–CH_2_), 21.3 (CH_2_–CH_2_), 18.2 (CH_2_–CH_2_), 13.4 (CH_2_–CH_3_) ppm. ^19^F NMR (376 MHz, DMSO-*d_6_*) δ −114.37 ppm. HRMS-ESI (*m/z*) [M+H]^+^ calcd. for C_21_H_18_FN_4_O_2_: 377.1408, found: 377.1404. λ_abs_: 273nm λ_em_: 498 nm λ_exc_: 286 nm. R*_f_* 0.33 (DCM/MeOH 95:5).

3-(*p*-Fluorophenyl)-8-hept-1-ynyl-*1H*-purino [9,8-*a*]pyridine-2,4-dione (**21d**). The title compound was prepared from commercially available **19b** to afford after purification the desired product as an orange solid (50%). ^1^H NMR (250 MHz, DMSO-*d_6_*) δ 8.56 (s, 1H, H_9_), 7.93–7.79 (m, 1H, H_6_), 7.57–7.41 (m, 1H, H_7_), 7.24 (dd, 4H, *J* = 28.7, 8.3 Hz, 4 × H_arom_), 2.60–2.45 (m, 2H, C–CH_2_), 1.58 (t, 2H, *J* = 7.2 Hz, CH_2_–CH_2_), 1.52–1.30 (m, 4H, 2 × CH_2_), 0.88 (dt, 3H, *J* = 13.5, 6.5 Hz, CH_2_–CH_3_) ppm. ^13^C NMR (63 MHz, DMSO-*d_6_*) δ 163.4 (C=O), 158.8 (C_quat_), 151.7 (C=O), 146.8 (C_quat_), 140.0 (C_quat_), 133.1 (d, *J_C–F_* = 3.2 Hz, C_quat_), 131.3 (d, *J_C–F_* = 11.7 Hz, 2 × C_arom_), 128.4 (C_arom_), 126.2 (C_arom_), 118.2 (C_arom_), 118.1 (C_quat_), 115.5 (d, *J_C–F_* = 24.3 Hz, 2 × C_arom_), 108.3 (C_quat_), 92.31 (CH_2_–C), 76.7 (C≡C), 30.4 (CH_2_–CH_2_), 27.6 (CH_2_–CH_2_), 21.5 (CH_2_–CH_2_), 18.6 (CH_2_–CH_2_), 13.8 (CH_2_–CH_3_) ppm. ^19^F NMR (376 MHz, DMSO-*d_6_*) δ −114.97 ppm. HRMS-ESI (*m/z*) [M+H]^+^ calcd. for C_22_H_20_FN_4_O_2_: 391.1564, found: 391.1563. λ_abs_: 271nm λ_em_: 515 nm λ_exc_: 284 nm. R*_f_* 0.33 (DCM/MeOH 95:5).

3-(*p*-Fluorophenyl)-8-oct-1-ynyl-*1H*-purino [9,8-*a*]pyridine-2,4-dione (**21e**). The title compound was prepared from commercially available 19b to afford after purification the desired product as an orange solid (74%). ^1^H NMR (400 MHz, DMSO-*d_6_*) δ 12.79 (s, 1H, H_1_), 8.63 (s, 1H, H_9_), 7.57 (d, 1H, *J* = 9.6 Hz, H_6_), 7.41–7.28 (m, 4H, 4 × H_arom_), 7.23 (dd, 1H, *J* = 9.6, 1.6 Hz, C–CH_2_), 2.48–2.42 (m, 2H, CH_2_–CH_2_), 1.62–1.52 (m, 2H, CH_2_–CH_2_), 1.48–1.38 (m, 2H, CH_2_–CH_2_), 1.32 (dq, 4H, *J* = 7.1, 3.3 Hz, 2 × CH_2_), 0.95–0.79 (m, 3H, CH_2_–CH_3_) ppm. ^13^C NMR (101 MHz, DMSO-*d_6_*) δ 162.7 (C=O), 160.3 (C_quat_), 158.4 (C_quat_), 150.2 (C=O), 140.3 (C_quat_), 132.3 (d, *J* = 4.3 Hz, C_quat_), 131.2 (d, *J_C–F_* = 9.9 Hz, 2 × C_arom_), 128.7 (C_arom_), 126.4 (C_arom_), 118.5 (C_quat_), 118.3 (C_arom_), 115.6 (d, *J_C–F_* = 22.6 Hz, 2 × C_arom_), 109.2 (C_quat_), 92.9 (CH_2_–C), 76.4 (C≡C), 30.7 (CH_2_–CH_2_), 27.9 (CH_2_–CH_2_), 27.9 (CH_2_–CH_2_), 21.9 (CH_2_–CH_2_), 18.5 (CH_2_–CH_2_), 13.9 (CH_2_–CH_3_) ppm. ^19^F NMR (376 MHz, DMSO-*d_6_*) δ −114.34 ppm. HRMS-ESI (*m/z*) [M+H]^+^ calcd. for C_23_H_22_FN_4_O_2_: 405.1721, found: 405.1720. R*_f_* 0.33 (DCM/MeOH 95:5). λ_abs_: 275 nm λ_em_: 503 nm λ_exc_: 286 nm.

*N*-[3-[3-(*p*-fluorophenyl)-2,4-dioxo-*1H*-purino [9,8-*a*]pyridin-8-yl]prop-2-ynyl]octanamide (**21f**). The title compound was prepared from commercially available 19b to afford after purification the desired product as an orange solid (40%). ^1^H NMR (400 MHz, DMSO-*d*_6_) δ 8.49 (s, 1H, H_9_), 7.39 (d, 1H,*J* = 9.4 Hz, H_6_), 7.28–7.17 (m, 4H, 4 × H_arom_), 7.06 (d, 1H, *J* = 9.4 Hz, H_7_), 4.07 (s, 1H, NH–CH_2_), 3.11–2.87 (m, 2H, CO–CH_2_), 1.43–1.28 (m, 2H, CH_2_–CH_2_), 1.21 (bs, 8H, 4 × CH_2_), 0.84 (t, 3H, *J* = 6.8 Hz, CH_2_–CH_3_) ppm. ^13^C NMR (63 MHz, DMSO-*d*_6_) δ 175.0 (C=O), 162.0 (C=O), 159.5 (C_quat_), 150.4 (C=O), 139.6 (C_quat_), 134.9 (C_quat_), 134.4 (C_quat_), 131.2 (d, *J_C–F_* = 16.8 Hz, 2 × C_arom_), 127.4 (C_arom_), 126.9 (C_arom_), 118.2 (C_arom_), 117.5 (C_quat_), 115.2 (d, *J_C–F_* = 14.4 Hz, 2 × C_arom_), 106.4 (C_quat_), 89.7 (CH_2_–C), 77.8 (C≡C), 45.4 (N–CH_2_), 35.0 (C–CH_2_), 31.0 (CH_2_–CH_2_), 29.9 (CH_2_–CH_2_), 29.5 (CH_2_–CH_2_), 26.0 (CH_2_–CH_2_), 22.0 (CH_2_–CH_2_), 13.9 (CH_2_–CH_3_) ppm. ^19^F NMR (376 MHz, DMSO-*d_6_*) δ −115.90 ppm. HRMS-ESI (*m/z*) [M+H]^+^ calcd. for C_26_H_27_FN_5_O_2_: 476.2092, found: 476.2091. λ_abs_: 273 nm λ_em_: 515 nm λ_exc_: 284 nm. R*_f_* 0.32 (DCM/MeOH 95:5).

1-[3-[3-(*p*-Fluorophenyl)-2,4-dioxo-*1H*-purino [9,8-*a*]pyridin-8-yl]prop-2-ynyl]-3-hexyl-urea (**21g**). The title compound was prepared from commercially available 19b to afford after purification the desired product as an orange solid (92%). ^1^H NMR (400 MHz, DMSO-*d_6_*) δ 8.49 (s, 1H, H_9_), 7.39 (d, 1H, *J* = 9.5 Hz, H_6_), 7.31–7.16 (m, 4H, 4 × H_arom_), 7.06 (d, 1H, *J* = 9.5 Hz, H_7_), 4.07 (d, 2H, *J* = 5.7 Hz, CH_2_), 1.45–1.03 (m, 10H, 5 × CH_2_), 0.84 (q, 3H, *J* = 5.7 Hz, CH_3_) ppm. ^13^C NMR (101 MHz, DMSO-*d_6_*) δ 162.0 (C=O), 159.9(C_quat_), 157.58 (C=O), 150.4 (C=O), 139.6 (C_quat_), 134.8 (C_quat_), 131.3 (C_quat_), 131.1 (2 × C_arom_), 127.3 (C_arom_),126.9 (C_arom_), 118.2 (C_arom_), 117.74 (C_quat_), 115.1 (2 × C_arom_), 106.4 (C_quat_), 89.7 (CH_2_–C), 77.8 (C≡C), 45.4 (N–CH_2_), 31.0 (CH_2_–CH_2_), 29.9 (CH_2_–CH_2_), 28.5 (CH_2_–CH_2_), 26.0 (CH_2_–CH_2_), 22.0 (CH_2_–CH_2_), 14.2 (CH_2_–CH_3_) ppm. ^19^F NMR (376 MHz, DMSO-*d_6_*) δ −116.17 ppm. HRMS-ESI (*m/z*) [M+H]^+^ calcd. for C_25_H_26_FN_6_O_3_: 477.2045, found: 477.2045. λ_abs_: 285 nm λ_em_: 512 nm λ_exc_: 286 nm. R*_f_* 0.30 (DCM/MeOH 95:5).

### 3.6. General Synthetic Procedure 1 for the Suzuki-Miyaura Coupling cross Coupling

A mixture of Br-compound (1 eq.), boronic acid (1.5 eq.), cesium carbonate (2 eq.), Pd(PPh_3_)_4_ (0.1 eq.) in anhydrous DMF (0.1 M) under inert atmosphere was heated under microwave irradiation at 120 °C for 40 min. Pure compounds were obtained after purification by flash column chromatography with DCM/MeOH as eluent.

3-(*p*-Fluorophenyl)-7-phenyl-*1H*-purino [9,8-*a*]pyridine-2,4-dione (**22a**). The title compound was prepared from commercially available 19a to afford after purification the desired product as a yellow solid (45%). ^1^H NMR (250 MHz, DMSO-*d*_6_) δ 8.52 (d, 1H, *J* = 7.3 Hz, H_9_), 7.85 (d, 3H, *J* = 6.9 Hz, H_6_, 2 × H_arom_), 7.60–7.34 (m, 4H, H_8_, 3 × H_arom_), 7.30 (d, 4H, *J* = 7.1 Hz, 4 × H_arom-N3_) ppm. ^13^C NMR (63 MHz, DMSO-*d*_6_) δ 163.1 (C=O), 159.2 (C_quat_), 159.0 (C_quat_), 152.1 (C=O), 142.0 (C_quat_), 137.2 (d, *J* = 14.2 Hz, C_quat_), 135.7 (C_quat_), 133.3 (d, *J_C–F_* = 9.3 Hz, C_quat_), 131.2 (d, *J_C–F_* = 9.1 Hz, 2 × C_arom-N3_), 128.9 (2 × C_arom_), 128.4 (C_arom_), 126.3 (2 × C_arom_), 123.9 (C_arom_), 118.4 (C_quat_), 115.4 (d, *J_C–F_* = 24.2 Hz, 2 × C_arom-N3_), 114.1 (C_arom_), 111.7 (C_arom_) ppm. ^19^F NMR (376 MHz, DMSO-*d_6_*) δ −115.15 ppm. HRMS-ESI (*m/z*) [M+H]^+^ calcd. for C_21_H_14_FN_4_O_2_: 373.1095, found: 373.1099. λ_abs_: 264 nm λ_em_: 517 nm λ_exc_: 285 nm. R*_f_* 0.31 (DCM/MeOH 95:5).

3-(*p*-Fluorophenyl)-7-(*p*-tolyl)-*1H*-purino [9,8-*a*]pyridine-2,4-dione (**22b**). The title compound was prepared from commercially available 19a to afford after purification the desired product as an orange solid (44%). ^1^H NMR (250 MHz, DMSO-*d*_6_) δ 8.33 (d, 1H, *J* = 7.3 Hz, H_9_), 7.72 (d, 2H, *J* = 8.1 Hz, 2 × H_arom_), 7.67 (s, 1H, H_6_), 7.31 (d, 2H, *J* = 8.0 Hz, 2 × H_arom_), 7.26–7.17 (m, 5H, H_8_, 4 × H_arom-N3_), 2.37 (s, 3H, C–CH_3_) ppm. ^13^C NMR (63 MHz, DMSO-*d*_6_) δ 159.9 (C=O), 158.7 (C_quat_), 152.3 (C=O), 141.5 (C_quat_), 138.0 (C_quat_), 137.7 (C_quat_), 136.3 (C_quat_), 135.1 (d, *J_C–F_ =* 2.5 Hz, C_quat_), 134.9 (C_quat_), 131.2 (d, *J_C–F_ =* 8.8 Hz, 2 × C_arom-N3_), 129.6 (2 × C_arom_), 126.4 (C_quat_), 126.1 (2 × C_arom_), 123.8 (C_arom_), 115.0 (d, *J_C–F_ =* 22.6 Hz, 2 × C_arom-N3_), 113.5 (C_arom_), 110.3 (C_arom_), 20.7 (C–CH_3_) ppm. ^19^F NMR (376 MHz, DMSO-*d_6_*) δ −116.31 ppm. HRMS-ESI (*m/z*) [M+H]^+^ calcd. for C_22_H_16_FN_4_O_2_: 387.1251, found: 387.1255. λ_abs_: 276 nm λ_em_: 519 nm λ_exc_: 279 nm. R*_f_* 0.32 (DCM/MeOH 95:5).

3-(*p*-Fluorophenyl)-7-(*p*-methoxyphenyl)-*1H*-purino [9,8-*a*]pyridine-2,4-dione (**22c**). The title compound was prepared from commercially available 19a to afford after purification the desired product as a yellow solid (15%). ^1^H NMR (250 MHz, DMSO-*d*_6_) δ 8.50 (d, 1H, *J* = 7.2 Hz, H_9_), 7.81 (d, 2H, *J* = 8.4 Hz, 2 × H_arom_), 7.77 (s, 1H, H_6_), 7.40–7.06 (m, 5H, H_8_, 4 × H_arom-N3_), 7.07 (d, 2H, *J* = 8.4 Hz, 2 × H_arom_), 3.82 (s, 3H, O–CH_3_) ppm. ^13^C NMR (63 MHz, DMSO-*d*_6_) δ 159.6 (C=O), 159.1 (C_quat_), 152.7 (C=O), 142.4 (C_quat_), 140.1 (C_quat_), 136.0 (C_quat_), 133.9 (C_quat_), 133.6 (d, *J_C–F_ =* 4.4 Hz, C_quat_), 131.2 (d, *J_C–F_ =* 8.19 Hz, 2 × C_arom-N3_), 129.6 (C_quat_), 127.7 (2 × C_arom_), 123.8 (C_arom_), 118.2 (C_quat_), 115.3 (d, *J_C–F_ =* 22.6 Hz, 2 × C_arom-N3_), 114.4 (2 × C_arom_), 112.8 (C_arom_), 111.5 (C_arom_), 55.2 (O–CH_3_) ppm. ^19^F NMR (376 MHz, DMSO-*d_6_*) δ −115.28 ppm. HRMS-ESI (*m/z*) [M+H]^+^ calcd. for C_22_H_16_FN_4_O_3_: 403.1200, found: 403.1202. λ_abs_: 275 nm λ_em_: 522 nm λ_exc_: 278 nm. R*_f_* 0.32 (DCM/MeOH 95:5).

3-(*p*-fluorophenyl)-7-[*p*-(trifluoromethoxy)phenyl]-*1H*-purino [9,8-*a*]pyridine-2,4-dione (**22d**). The title compound was prepared from commercially available 19a to afford after purification the desired product as an orange solid (15%). ^1^H NMR (250 MHz, DMSO-*d*_6_) δ 8.53 (d, 1H, *J* = 7.4 Hz, H_9_), 7.98 (d, 2H, *J* = 8.5 Hz, 2 × H_arom_), 7.88 (s, 1H, H_6_), 7.49 (d, 2H, *J* = 8.5 Hz, 2 × H_arom_), 7.39 (d, 1H, *J* = 7.4 Hz, H_8_), 7.30 (d, 4H, *J* = 7.0 Hz, 4 × H_arom-N3_) ppm. ^13^C NMR (63 MHz, DMSO-*d*_6_) δ 163.1 (C=O), 159.3 (C_quat_), 159.0 (C_quat_), 152.2 (C=O), 148.4 (C_quat_), 141.7 (C_quat_), 136.7 (C_quat_), 136.1 (C_quat_), 135.6 (C_quat_), 133.4 (C_quat_), 131.2 (d, *J_C–F_* = 8.7 Hz, 2 × C_arom-N3_), 128.5 (2 × C_arom_), 124.1 (C_arom_), 121.4 (2 × C_arom_), 118.6 (C_quat_), 115.3 (d, *J_C–F_* = 22.6 Hz, 2 × C_arom-N3_), 114.8 (C_arom_), 111.6 (C_arom_) ppm. ^19^F NMR (376 MHz, DMSO-*d_6_*) δ –56.71 (–OCF_3_), −115.18 ppm. HRMS-ESI (*m/z*) [M+H]^+^ calcd. for C_22_H_13_F_4_N_4_O_3_: 457.0918, found: 457.0917. λ_abs_: 274 nm λ_em_: 536 nm λ_exc_: 279 nm. R*_f_* 0.3 (DCM/MeOH 95:5).

3,7-*Bis*(*p*-fluorophenyl)-*1H*-purino [9,8-*a*]pyridine-2,4-dione (**22e**). The title compound was prepared from commercially available **19a** to afford after purification the desired product as a yellow solid (45%). ^1^H NMR (250 MHz, DMSO-*d*_6_) δ 8.50 (d, 1H, *J* = 7.4 Hz, H_9_), 7.97–7.86 (m, 2H, 2 × H_arom-N3_), 7.82 (s, 1H, H_6_), 7.44–7.22 (m, 7H, H_8_, 2 × H_arom-N3_, 4 × H_arom-N3_) ppm. ^13^C NMR (63 MHz, DMSO-*d*_6_) δ 160.3 (C=O), 159.1 (C_quat_), 159.0 (C_quat_), 152.4–152.1 (m, C=O, C_quat_), 141.8 (C_quat_), 136.0 (C_quat_), 133.8 (d, *J_C–F_* = 3.1 Hz, C_quat_), 133.5 (C_quat_), 131.2 (d, *J_C–F_* = 8.8 Hz, 2 × C_arom-N3_), 128.6 (d, *J_C–F_* = 8.1 Hz, 2 × C_arom_), 124.5 (C_quat_), 124.0 (C_arom_), 118.4 (C_quat_), 115.8 (d, *J_C–F_* = 21.4 Hz, 2 × C_arom-N3_), 115.3 (d, *J_C–F_* = 22.6 Hz, 2 × C_arom_), 114.1 (C_arom_), 111.6 (C_arom_) ppm. ^19^F NMR (235 MHz, DMSO-*d*_6_) δ −113.66, −115.26 ppm. HRMS-ESI (*m/z*) [M+H]^+^ calcd. for C_21_H_13_F_2_N_4_O_2_: 391.1002, found: 391.1001. λ_abs_: 274 nm λ_em_: 539 nm λ_exc_: 277 nm. R*_f_* 0.26 (DCM/MeOH 95:5).

3-(*p*-Fluorophenyl)-8-phenyl-*1H*-purino [9,8-*a*]pyridine-2,4-dione (**23a**). The title compound was prepared from commercially available 19b to afford after purification the desired product as a yellow solid (50%). ^1^H NMR (400 MHz, DMSO-*d*_6_) δ 8.86 (s, 1H, H_9_), 7.75–7.72 (m, 2H, 2 × H_arom_), 7.67 (qd, 2H, *J* = 9.7, 1.4 Hz, H_6_, H_arom_), 7.54 (t, 2H, *J* = 7.7 Hz, 2 × H_arom_), 7.47–7.41 (m, 1H, H_7_), 7.36–7.27 (m, 4H, 4 × H_arom-N3_) ppm. ^13^C NMR (101 MHz, DMSO-*d*_6_) δ 162.5 (C=O), 160.1 (C_quat_), 158.9 (C_quat_), 151.5 (C=O), 140.8 (C_quat_), 136.0 (C_quat_), 134.0 (C_quat_), 133.1 (d, *J_C–F_* = 2.8 Hz, C_quat_), 131.3 (d, *J_C–F_* = 9.1 Hz, 2 × C_arom-N3_), 129.2 (2 × C_arom_), 128.1 (C_arom_), 126.2 (2 × C_arom_), 125.2 (C_quat_), 120.9 (C_arom_), 118.3 (C_arom_), 115.4 (d, *J_C–F_* = 22.7 Hz, 2 × C_arom-N3_) ppm. ^19^F NMR (376 MHz, DMSO-*d_6_*) δ −114.90 ppm. HRMS-ESI (*m/z*) [M+H]^+^ calcd. for C_21_H_14_FN_4_O_2_: 373.1095, found: 373.1094. λ_abs_: 274 nm λ_em_: 441 nm λ_exc_: 288 nm. R*_f_* 0.31 (DCM/MeOH 95:5).

3-(*p*-Fluorophenyl)-8-(*p*-tolyl)-*1H*-purino [9,8-*a*]pyridine-2,4-dione (**23b**). The title compound was prepared from commercially available **19b** to afford after purification the desired product as a yellow/beige solid (31%). ^1^H NMR (250 MHz, DMSO-*d*_6_) δ 8.82 (s, 1H, H_9_), 7.62 (d, 5H, *J* = 7.9 Hz, H_6,_ 4 × H_arom_), 7.33 (t, 7H, *J* = 6.7 Hz, H_7_, 2 × H_arom_, 4 × H_arom-N3_), 2.37 (s, 3H, C–CH_3_) ppm. ^13^C NMR (63 MHz, DMSO-*d*_6_) δ 158.9 (C=O), 157.0 (C_quat_), 151.6 (C=O), 150.8 (C_quat_), 140.8 (C_quat_), 137.5 (C_quat_), 135.0 (C_quat_), 133.1 (d, *J_C–F_* = 3.7 Hz, C_quat_), 133.0 (C_quat_), 131.3 (d, *J_C–F_* = 8.8 Hz, 2 × C_arom-N3_), 129.7 (2 × C_arom_), 126.2 (C_arom_), 126.0 (2 × C_arom_), 125.0 (C_quat_), 120.4 (C_arom_), 118.2 (C_arom_), 115.4 (d, *J_C–F_* = 22.6 Hz, 2 × C_arom-N3_), 20.6 (C–CH_3_) ppm. ^19^F NMR (376 MHz, DMSO-*d_6_*) δ −114.93 ppm. HRMS-ESI (*m/z*) [M+H]^+^ calcd. for C_22_H_16_FN_4_O_2_: 387.1251, found: 387.1253. λ_abs_: 244 nm λ_em_: 444 nm λ_exc_: 274 nm. R*_f_* 0.32 (DCM/MeOH 95:5).

3-(*p*-Fluorophenyl)-8-(*p*-methoxyphenyl)-*1H*-purino [9,8-*a*]pyridine-2,4-dione (**23c**). The title compound was prepared from commercially available 19b to afford after purification the desired product as a yellow/beige solid (31%). ^1^H NMR (400 MHz, DMSO-*d*_6_) δ 8.78 (s, 1H, H_9_), 7.70–7.57 (m, 4H, H_6_, H_7_, 2 × H_arom_), 7.33–7.30 (m, 4H, 4 × H_arom-N3_), 7.10 (d, 2H, *J* = 8.7 Hz, 2 × H_arom_), 3.82 (s, 3H, O–CH_3_) ppm. ^13^C NMR (101 MHz, DMSO-*d*_6_) δ 162.5 (C=O), 162.5 (C_quat_), 159.3 (C_quat_), 151.6 (C=O), 140.7 (C_quat_), 140.0 (C_quat_), 133.1 (C_quat_), 131.3 (d, *J_C–F_ =* 9.0 Hz, 2 × C_arom-N3_), 128.2 (C_quat_), 127.4 (2 × C_arom_), 126.3 (C_arom_), 124.9 (C_quat_), 119.9 (C_arom_), 118.2 (C_arom_), 115.4 (d, *J_C–F_ =* 22.2 Hz, 2 × C_arom-N3_), 114.6 (2 × C_arom_), 99.4 (C_quat_), 55.2 (O–CH_3_) ppm. ^19^F NMR (376 MHz, DMSO-*d_6_*) δ −114.96 ppm. HRMS-ESI (*m/z*) [M+H]^+^ calcd. for C_22_H_16_FN_4_O_3_: 403.1200, found: 403.1201. λ_abs_: 274 nm λ_em_: 438 nm λ_exc_: 286 nm. R*_f_* 0.32 (DCM/MeOH 95:5).

3-(*p*-Fluorophenyl)-8-[*p*-(trifluoromethoxy)phenyl]-*1H*-purino [9,8-*a*]pyridine-2,4-dione (**23d**). The title compound was prepared from commercially available **19b** to afford after purification the desired product as a yellow/beige solid (43%). ^1^H NMR (250 MHz, DMSO-*d*_6_) δ 8.68 (s, 1H, H_9_), 7.94–7.84 (m, 2H, 2 × H_arom_), 7.48 (d, 1H, *J* = 9.7 Hz, H_6_), 7.39–7.22 (m, 7H, H_7_, 2 × H_arom_, 4 × H_arom-N3_) ppm. ^13^C NMR (101 MHz, DMSO-*d*_6_) δ 164.6 (C=O), 159.3 (C_quat_), 159.0 (C_quat_), 152.5 (C=O), 146.2 (C_quat_), 139.7 (C_quat_), 138.9 (C_quat_), 136.1 (2 × C_arom_), 133.3 (C_quat_), 131.2 (d, *J_C–F_* = 3.1 Hz, C_quat_), 131.9 (C_quat_), 131.2 (d, *J_C–F_* = 8.8 Hz, 2 × C_arom_), 128.7 (C_arom_), 124.0 (C_arom_), 119.5 (d, *J_C–F_* = 5.5 Hz, 2 × C_arom_), 118.3 (C_quat_), 115.3 (d, *J_C–F_* = 22.8 Hz, 2 × C_arom_), 106.0 (C_quat_) ppm. ^19^F NMR (376 MHz, DMSO-*d_6_*) δ –56.72 (–O–CF_3_), −114.27 ppm. HRMS-ESI (*m/z*) [M+H]^+^ calcd. for C_22_H_13_F_4_N_4_O_3_: 457.0919, found: 457.0919. λ_abs_: 273 nm λ_em_: 510 nm λ_exc_: 294 nm. R*_f_* 0.3 (DCM/MeOH 95:5).

3,8-Bis(p-Fluorophenyl)-1H-purino [9,8-*a*]pyridine-2,4-dione (**23e**). The title compound was prepared from commercially available 19b to afford after purification the desired product as a yellow/beige solid (11%). ^1^H NMR (400 MHz, DMSO-*d_6_*) δ 8.90 (s, 1H, H_9_), 7.82–7.73 (m, 2H, 2 × H_arom_), 7.69–7.58 (m, 2H, H_6_, H_7_), 7.38 (t, 2H, *J* = 8.8 Hz, 2 × H_arom_), 7.32–7.22 (m, 4H, 4 × H_arom-N3_) ppm. ^13^C NMR (101 MHz, DMSO-*d_6_*) δ 160.8 (C=O), 160.0 (C_quat_), 159.1 (C_quat_), 151.9 (C=O), 140.7 (C_quat_), 136.4 (C_quat_), 135.9 (C_quat_), 132.5 (d, *J_C–F_* = 3.5 Hz, C_quat_), 131.5–13.4 (m, C_quat_), 131.3 (d, *J_C–F_* = 8.4 Hz, 2 × C_arom_), 128.4 (d, *J_C–F_* = 8.3 Hz, 2 × C_arom-N3_), 126.1 (C_arom_), 124.0 (C_quat_), 121.0 (C_arom_), 118.3 (C_arom_), 116.0 (d, *J_C–F_* = 21.8 Hz, 2 × C_arom_), 115.4 (d, *J_C–F_* = 22.7 Hz, 2 × C_arom-N3_) ppm. ^19^F NMR (376 MHz, DMSO-*d_6_*) δ −114.36, −115.10 ppm. HRMS-ESI (m/z) [M+H]^+^ calcd. for C_21_H_13_F_2_N_4_O_2_: 391.1001, found: 391.1006. λ_abs_: 274 nm λ_em_: 441 nm λ_exc_: 273 nm. R*_f_* 0.26 (DCM/MeOH 95:5).

### 3.7. Biological Assays

#### 3.7.1. Mtb ThyX Protein Expression and Purification [30]

The *M. tuberculosis* ThyX enzyme was expressed in *E. coli* BL21(DE3)/pLysS strains containing the recombinant pET24d plasmid carrying the *M. tuberculosis* H37Rv thyX gene (Rv2754c) as previously described. Before the purification step, 200 μM of flavin-adenine dinucleotide (FAD) cofactor was added to the supernatant after the lysis step to increase the amount of FAD bound to *Mtb* ThyX protein. The solubilized protein extract was loaded on a Hi-Trap Talon 5 mL column (GE Healthcare) previously equilibrated with equilibration buffer containing 30 mM Hepes and 300 mM NaCl at pH 8.0. The His-tagged ThyX protein was eluted with elution buffer (30 mM Hepes pH 8.0, 300 mM NaCl, 500 mM imidazol). The fractions containing Mtb ThyX enzyme were pooled, buffer-exchanged on Econo-Pac PD-10 columns (Bio-rad) with the equilibration buffer, concentrated to a final concentration of 480 μM and stored at −20 °C for further use. The measured absorbance of FAD bound to *Mtb* ThyX at 450 nm showed a ratio FAD to ThyX of 1 to 3 for the purified *Mtb* ThyX chain.

#### 3.7.2. M. tuberculosis ThyX NADPH Oxidase Assay

The NADPH oxidation assay for *M. tuberculosis* ThyX activity in 96-well plates was used to screen the synthesized compounds at a final concentration of 200 μM. All molecules were solubilized in dimethylsulfoxide (DMSO) and used at a 1% final concentration of DMSO during the test. One hundred microlitres of standard reaction mixture contained HEPES 50 mM pH 8, NaCl 30 mM, FAD 50 μM, β-mercaptoethanol 1.43 mM, dUMP 100 μM, NADPH 750 μM, and 10 μM of purified *Mtb*ThyX. Microtiter plates were prepared and transferred to the microplate reader Chameleon II (Hidex). Molecules at 200 μM were incubated with *Mtb*ThyX in the standard reaction mixture for 10 min at 25 °C before starting measurements. The reactions were started by automatically injecting NADPH into individual wells and ThyX activity was determined by following a decrease in absorbance at 340 nm for up to 20 min at 25 °C. The experiment was done in duplicates and samples with added DMSO and enzyme-free reactions were used as positive and negative controls, respectively. % of inhibition was calculated using the following equation: (1 − Vi/Vo)*100; Vo and Vi are, respectively, the initial rates of the reaction without or with addition of molecule in the assay.

#### 3.7.3. M. tuberculosis ThyX Tritium Release Assays

Mtb ThyX tritium release assays consist to measure “deprotonation” of [5-^3^H]-dUMP in vitro. Reaction mixture included 10 mM MgCl_2_, 500 µM FAD, 10% (*v*/*v*) glycerol, 2 mM NADPH, 1 mM CH_2_H_4_folate*, 10 mM* β*-mercaptoethanol,* bovine serum albumin (200 µg/mL), 100 µM dUMP, [5*-^3^H*]dUMP and *2* µM Mtb ThyX in 50 mM HEPES pH 8. Molecules at 200 μM *in DMSO* were incubated with *Mtb*ThyX in the standard reaction mixture for 10 min at 25 °C before starting measurements. DMSO concentration was maintained constant at 1%. Reactions were initiated by addition of NADPH (1 mM) at 37 °C and were stopped after 20 min. The specific activity of tritiated [5-^3^*H*]dUMP (diammonium salt) stock was 15–30 Ci mmol^−1^ (Moravek Biochemicals, CA, USA). 700 μL of activated charcoal (10% (*w*/*v*), Norit A in 2% trichloroacetic acid) was added to the reaction mixture to stop the reaction and removal of radioactive nucleotides from the solution. The suspension was centrifuged at 12,000 rpm for 2 min, and 450 μL of the supernatant were collected before addition of 5 mL of scintillation solution. The radioactivity remaining in the supernatant was measured for scintillation counting.

#### 3.7.4. Cytotoxicity Assays

Assays were performed in human peripheral blood mononuclear (PBM) cells via MTS assay using the CellTiter 96^®^ Non-Radioactive Cell Proliferation (Promega) kit. Cytotoxicity was expressed as the concentration of test compounds that inhibited cell growth by 50% (CC_50_).

### 3.8. Virtual Docking

The MtbThyX protein structure from PDB code 3GWC26 was used to perform in silico molecular docking with the QuickVina 2 software [31]. The A to D chains, water oxygen atoms and UFP cofactor were removed from the structure, keeping only chains F to G and their FAD molecule. Atomic partial charges were assigned and polar hydrogen atoms were added with the Pymol [32] Vina plugin [33]. A cubic search volume of 35 × 35 × 35 Å centered on each active site of the four chains was used. The Arg87, Gln103, Ser105, Arg107, Tyr108 and Arg199 amino acids were chosen to be flexible during the docking attempts. Ten docking poses were generated and the pose with the best score was used to further analysis.

## 4. Conclusions

In summary, starting from lead compounds 5 and 6, we have synthesized various hitherto unknown heterocycles with a pyrido [1,2-*e*]purine-2,4(1*H*,3*H*)-dione scaffold. Structure-activity relationship was performed in order to guide the design of new *FDTS* inhibitors. Several pharmacomodulations through cylization, and regioselective *N*-alkylation and organopalladium cross-coupling reactions were performed to synthesize *N^3^, N^1^*, *C^7^* and *C^8^* alkylated derivatives. Unfortunately no compound showed total inhibition at low concentration; the highest inhibition was obtained for compound 23a with 84.3% at 200 µM. IC_50_ was calculated at 95 µM for central fluorine structure 9a. It was also evaluated on viral pathogens and showed some promiscuous activity without toxicity. Docking was performed to understand the main interactions of those compounds in *ThyX*. Altogether, this paves the route to more potent *Flavin-Dependent Thymidylate Synthase* inhibitors.

## Data Availability

The data present in study are available in the
Appendix A.

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
