# Peer review of "Synthesis and Structure–Activity Relationship Studies of Pyrido [1,2-e]Purine-2,4(1H,3H)-Dione Derivatives Targeting Flavin-Dependent Thymidylate Synthase in Mycobacterium tuberculosis"

_molecules, 2022, doi:10.3390/molecules27196216_

Round 1
Reviewer 1 Report
The authors report some pyrido[1,2-e]purine-2,4(1H,3H)-dione analogues as a Mycobacterium tuberculosis ThyX inhibitors. compound 23a is a potent one. If they can test the antituberculosis activity of 23a, this work will be more attractive. There are some errors in manuscript which need to be revised.
some comments as below: 1. In Figure 1, %inhib should be gave as %inhibition 2. line 85, multidrug-resistant strains (MDR) of Mycobacterium tuberculosis is multidrug-resistant tuberculosis (MDR-TB) 3. the structure of B1-PP146 should be gave 4. line 122, N3 should be italic 5. in scheme 2, the substituents of 9j don't be displayed, 11 R1=CF, what is CF? 6. in scheme 3, the main structure of 16 and 17 is wrong 7. the serial number of substituents should be consistent, in scheme 3, R3 and R4, however, in scheme 4, they changed to R1 and R2, and in Table 4, they chande to R3A and R4A. It is difficult to follow. The same question in scheme 5 and table 5. 8. in abstract, the author said 'Highest 24 inhibition was obtained for compound 23a with 84.3% at 200 μM with no or low toxicity,' , I don't find the results about toxicity.
Author Response
-> All suggested corrections have been done.
-> In experimental, a short paragraph was added to describe the cytotoxicity assay done on PBM cells. No significant cytotoxicity was observed (CC50 > 100 microM)
-> Compound 23a gives the best % of inhibition on FDTS of Mycobacterium Tuberculosis but at a tested concentration (200 microM) which remains too high and does not justify to measure directly the antituberculosis activity.
Reviewer 2 Report
The manuscript by Agrofoglio and co-workers describe the synthesis of pyrido[1,2-E]purine-2,4(1H,3H)-dione derivatives and their SAR studies. They synthesized various derivatives and SAR studies very well. I find this submitted manuscript is very intriguing from a biological perspective, also synthetic chemistry in this submitted manuscript is quite solid, synthesized compounds are well characterized. I would recommend its publication in Molecules if the authors can kindly address the following comments.
Minor comments:
Please check the compound numbers in Scheme 2. Compound numbers should be in bold character.
Please present all chemical structures of reagents in Scheme 2.
Please check “alkyn chain” in Scheme 4. It is corrected to “alkynyl chain”
Author Response
-> All suggested corrections have been done.